# You Don't Need All That Attention:
# Surgical Memorization Mitigation in Text-to-Image Diffusion Models

**Kairan Zhao** [1]  **Eleni Triantafillou** [2]  **Peter Triantafillou** [1]

## Abstract

Generative models have been shown to "memorize" certain training data, leading to verbatim or near-verbatim generating images, which may cause privacy concerns or copyright infringement. We introduce Guidance Using Attractive-Repulsive Dynamics (GUARD), a novel framework for memorization mitigation in text-to-image diffusion models. GUARD adjusts the image denoising process to guide the generation away from an original training image and towards one that is distinct from training data while remaining aligned with the prompt, guarding against reproducing training data, without hurting image generation quality. We propose a concrete instantiation of this framework, where the positive target that we steer towards is given by a novel method for (cross) attention attenuation based on (i) a novel statistical mechanism that automatically identifies the prompt positions where cross attention must be attenuated and (ii) attenuating cross-attention in these per-prompt locations. The resulting GUARD offers a surgical, dynamic per-prompt inference-time approach that, we find, is by far the most robust method in terms of consistently producing state-of-the-art results for memorization mitigation across two architectures and for both verbatim and template memorization, while also improving upon or yielding comparable results in terms of image quality.

## 1. Introduction

Researchers have recently observed that generative models can reproduce verbatim or near-verbatim copies of certain training examples (Carlini et al., 2021; 2023; Somepalli et al., 2023a); a phenomenon referred to as "memorization".

This is problematic for various reasons, ranging from revealing sensitive or private data to potential copyright concerns, and numerous research efforts have attempted to address the issue of memorization either preventatively or post-hoc. We study this important problem in the context of text-to-image (T2I) diffusion models.

Several approaches have been proposed to address memorization, including "training time" methods that act during the original training phase, to prevent memorization from occurring in the first place (Ren et al., 2024; Wen et al., 2024), and methods operating at post hoc, "finetuning time" that aim to post-process a model to remove the memorization of certain examples ("unlearning") (Golatkar et al., 2020; Kurmanji et al., 2023; Shah et al., 2024; Fan et al., 2024; Zhao et al., 2024; Alberti et al., 2025). However, these methods have various drawbacks: in modern pipelines, practitioners often build on pretrained models and lack control of the original pretraining phase, making training-time interventions unrealistic. Even setting that unrealistic assumption aside, training-time interventions that can at best guess which examples will become memorized, rely on heuristics and may have unwanted side-effects such as deteriorating the model's utility. Finetuning-based unlearning methods, on the other hand, are both computationally inefficient and often lack robustness (Łucki et al., 2024; Siddiqui et al., 2025).

Given the above shortcomings of training- and finetuning-time methods, and the desiderata of computational efficiency and surgical precision of memorization reduction without compromising utility, we turn our attention to a third category of inference-time mitigation: rather than seeking a model that has no memorized information in its weights, we accept that memorization may have occurred, but devise specialized inference-time procedures that mask memorized information so that it doesn't affect the generated content. In other words, while some memorized information may be present in the model weights, we seek a specialized forward pass at inference-time that: (i) guards against generating any memorized training example; (ii) does so in a manner that does not adversely affect the high-quality of generated images; and (iii) with minimal impact to efficiency.

Our first contribution is a novel inference-time mitigation framework for T2I diffusion models called Guidance Us-

---

[1]University of Warwick, UK [2]Google DeepMind, UK. Correspondence to: Kairan Zhao <Kairan.Zhao@warwick.ac.uk>.

*Proceedings of the $43^{rd}$ International Conference on Machine Learning*, Seoul, South Korea. PMLR 306, 2026. Copyright 2026 by the author(s).

ing Attractive-Repulsive Dynamics (GUARD). We build on the classic "classifier-free guidance" formula, that sets the predicted noise for a given timestep of denoising to be a combination of the predicted noise obtained from an empty prompt (the "unconditional prediction") and the predicted noise obtained from the given text prompt (the "text-conditional prediction", obtained via cross-attention with the text prompt embeddings). We modify this guidance to add a *negative* weight to the text-conditional prediction (the "repulsion" term), and we add an additional "attraction" term with a positive weight. Effectively performing arithmetic of the predicted noise, our revised "contrastive guidance" is governed by two forces. The "repulsive" term encourages moving away from the predicted noise that the standard forward pass would have produced on the given prompt. This is a key driver of memorization mitigation, since that standard forward pass applied on a memorized prompt would steer the generation towards reconstructing the training image associated with that prompt. At the same time, the "attraction term" provides a better target to steer towards. This term not only helps with mitigating memorization but acts as a quality booster, as steering aggressively away from the memorized prompt can otherwise cause fidelity collapse, where generated images lose structural coherence or semantic relevance to the prompt.

Next, we propose a concrete instantiation of the GUARD framework via a novel cross-attention adjustment method playing the role of the "positive target" that GUARD steers towards. We arrived at this method after a careful empirical analysis of the distribution of cross-attention at particular tokens, building and expanding on past work that found that specific "trigger" tokens are responsible for regurgitating training images (Somepalli et al., 2023a; Wen et al., 2024). Compared to prior work that uses hard-coded strategies to redistribute the (cross)-attention (Ren et al., 2024), we develop a dynamic on-the-fly and per-prompt trigger-token detection strategy to inform attention redistribution. We demonstrate that this approach improves upon the prior state of the art on its own, but even by a larger margin when integrated into GUARD as the positive target.

We summarize our key contributions below.

- **Framework:** We propose GUARD, a contrastive guidance framework for inference-time memorization mitigation that combines repulsion from memorized directions with attraction toward a safe target.

- **Empirical Analysis.** We analyze the distribution of cross attention at tokens of memorized and non-memorized prompts, across model architectures and for verbatim and template memorization, revealing more nuanced desiderata for memorization mitigation.

- **Detection:** Guided by our analysis, we design a per-

prompt cross-attention spike detector that identifies memorization-critical positions $\mathcal{S}(p)$ for a prompt $p$ based on on-the-fly statistical outlier analysis.

- **Instantiation:** We instantiate GUARD with a surgical cross-attention-logit attenuation mechanism that attenuates attention at positions in $\mathcal{S}(p)$.

- **Evaluation:** We conduct a comprehensive evaluation of GUARD and its instantiations versus several prior state-of-the-art methods. The results show that our method outperforms prior work across model architecture, memorization types (verbatim vs template) and according to both memorization and quality metrics.

## 2. Background

**Diffusion Models**. In image diffusion models, a "forward diffusion" process first adds Gaussian noise to an image. This happens gradually over $T$ steps, with each step adding a predetermined amount of noise. The image $x_t$ at step $t$ of this forward process can be computed in closed form as:

$$x_t = \sqrt{\bar{\alpha}_t} x_0 + \sqrt{1 - \bar{\alpha}_t}\epsilon, \tag{1}$$

where $x_0$ is the original image (before any noise addition), and $\bar{\alpha}_t = \prod_{i=1}^{t}(1 - \beta_t)$, where $\beta_t \in (0, 1)$ is the scheduled variance at step $t$, and $\epsilon \sim \mathcal{N}(0, I)$.

Then, the "reverse diffusion" process gradually removes the noise, aiming to obtain the original clean image. This happens via a trainable "noise predictor" model $\epsilon_\theta$, parameterized by $\theta$, that takes as input a noised image and predicts the noise that was added to it. The training objective for parameters $\theta$ is then the MSE loss between the sampled noise and the predicted noise across timesteps $t$:

$$\mathcal{L}(\theta) = \mathbb{E}_{t, x_0, \epsilon \sim \mathcal{N}(0,1)} \left\| \epsilon_t - \epsilon_\theta(x_t, t) \right\|_2^2 \tag{2}$$

**Text-to-Image Diffusion Models**. Some text-conditional diffusion models, such as Stable Diffusion, use classifier-free guidance (Rombach et al., 2022) at sampling time to steer generations. During inference, given a text prompt $p$, it is first embedded into a latent space using a text encoder $f$, to obtain the prompt embedding $e_p = f(p)$. The predicted noise at step $t$ is now

$$\epsilon_\theta(x_t, t, e_p) = \epsilon_\theta(x_t, t, e_\emptyset) + s(\epsilon_\theta(x_t, t, e_p) - \epsilon_\theta(x_t, t, e_\emptyset)) \tag{3}$$

where $e_\emptyset$ is the embedding of the empty prompt and $s$ represents the strength of the "text guidance", controlling how strongly the prediction should be influenced by the prompt.

In these models, typically the denoising network is a U-Net. A U-Net predicts a noise $\epsilon_\theta(x_t, t, e_p)$ from a noisy latent $x_t$ at timestep $t$, conditioned on the prompt embedding $e_p$. The

U-Net consists of a "downsampling" path, a "bottleneck" (mid) block, and an "upsampling" path, with skip connections between corresponding resolutions. Text conditioning is achieved via cross-attention layers placed at multiple U-Net blocks, enabling spatial latent features to selectively attend to text-token embeddings.

Concretely, let $H \in \mathbb{R}^{N \times d}$ denote the sequence of latent features at some U-Net block (e.g., $N$ spatial positions flattened) and let $E \in \mathbb{R}^{L \times d}$ denote the sequence of $L$ text token embeddings from the text encoder. A standard cross-attention layer forms queries $Q = HW_Q \in \mathbb{R}^{N \times d_k}$, keys $K = EW_K \in \mathbb{R}^{L \times d_k}$, and values $V = EW_V \in \mathbb{R}^{L \times d_v}$. Attention logits are $\ell = QK^\top / \sqrt{d_k} \in \mathbb{R}^{N \times L}$, and attention weights are $A = \mathrm{softmax}(\ell)$ over the token, producing attended features $AV$. In later sections we manipulate a subset of these logits $\ell_{q,i}$ for selected token indices $i$ (and queries $q$) prior to the softmax. This serves as a mechanism to reduce the weight of memorization-critical tokens, while leaving the rest intact.

**Memorization types**. Prior work identified and distinguished between two types of "memorization" in text-to-image diffusion models: "verbatim memorization" and "template memorization" (Webster, 2023; Ren et al., 2024). Verbatim memorization refers to the model reproducing near-identical copies of specific training instances when prompted with the original text. In contrast, template memorization manifests as the model reproducing images closely aligned with the training instances while allowing non-semantic variations. Although both phenomena reflect memorization, they arise from different mechanisms and exhibit distinct behavioral signatures. This is an important distinction as we will show later that prior methods designed for one type of memorization fail to mitigate the other type. We are after a general mitigation method that works across the board, so we design with this desideratum in mind.

## 3. Related Work

Related efforts can be categorized into three broad categories; we present an overview below.

**Training Time Memorization Mitigation (TTMM)** methods intervene during training to prevent memorization from occurring (Ren et al., 2024; Wen et al., 2024). If successful, no memorized information will be in the weights, making the resulting model robust against white-box extraction attacks. However, these methods are often "blunt instruments" that operate at a coarse granularity: because they operate during training when it's not clear which examples will become memorized, they may overly suppress example influence with detrimental consequences for learning or unintended side effects such as a utility degradation and the indiscriminate suppression of benign but rare visual concepts.

Furthermore, in modern pipelines, it is common practice to leverage pretrained models whose training process we have no control over, making TTMM unrealistic.

**Machine unlearning** methods aim to remove the influence of specific training data from models post-training, therefore naturally lending themselves to memorization mitigation. These methods do not assume control over the original training phase, overcoming that limitation of TTMM. Foundational unlearning algorithms were designed for image classification using techniques like NegGrad (Golatkar et al., 2020), random labeling (Graves et al., 2021), NegGrad+ (Kurmanji et al., 2023), SCRUB (Kurmanji et al., 2023), SalUn (Fan et al., 2024), L1-Sparse (Shah et al., 2024), and RUM (Zhao et al., 2024) and later adapted to LLMs (Yao et al., 2024; Bărbulescu & Triantafillou, 2024). In the domain of text-to-image (T2I) diffusion models (DMs), most existing unlearning research focuses on concept-level forgetting, which aims to erase broad visual categories or artistic styles rather than individual examples (Gandikota et al., 2023; Cywiński & Deja, 2025; Chen et al., 2025; Schioppa et al., 2024; Zhang et al., 2024; Ko et al., 2024; Park et al., 2024). Recently, Alberti et al. (2025) tackle the issue of example-level unlearning in text-to-image models, which is the most closely related to our work out of the unlearning methods. However, unlearning approaches come with several issues: they are computationally inefficient as they require a finetuning phase for each given forget set, and recent work shows that believed-to-be unlearned information often surfaces spontaneously, raising concerns about the effectiveness of these methods (Siddiqui et al., 2025; Hu et al., 2024; Deeb & Roger, 2024; Łucki et al., 2024).

**Inference Time Memorization Mitigation (ITMM).** Unlike the previous categories, ITMM methods do not modify the weights. Instead, they operate on the fly at inference time. ITMM is particularly attractive due to its expected computational efficiency and its ability to operate without access to the original training phase. Wen et al. (2024) discovered that memorized prompts produce abnormally large text-conditional noise prediction magnitudes and proposed scaling down the conditional signal during sampling. Furthermore, Ren et al. (2024) demonstrated that memorization is mechanistically linked to peaky Cross-Attention (CA) distributions on "trigger tokens," specifically summary tokens such as the end-ot-text (EOT) and padding tokens. They propose an inference-time memorization mitigation technique that redistributes cross-attention among tokens, indirectly. Specifically, before softmax normalization the method reduces cross-attention at the EOT and padding tokens (to $-\infty$) and boosts it at BOT (the beginning of text token). We contribute an analysis that explains why and when this method performs poorly. More recently, Han et al. (2025) attributed memorization to an "attraction basin" in the sample-time space where classifier-free guidance (CFG)

pulls trajectories toward memorized outputs; they proposed adjusting the initial noise sample to facilitate an earlier escape from this basin (Jain et al., 2025). Similarly, Jeon et al. (2025) offer a complementary geometric perspective, linking memorization in diffusion models to sharp regions of the learned probability landscape and proposing a sharpness-aware adjustment of the initial noise for mitigation. Our work augments related work by contributing a novel suite of surgical approaches to inference-time mitigation and comprehensively study their performance. While the EOT token is a primary summary shortcut, memorization is also driven by instance-specific trigger tokens *that do not follow consistent categorical patterns* (i.e. they cannot be identified solely through linguistic or semantic classifications—such as Part-of-Speech (POS) tags or named entity types, a fact also supported by the work in (Wen et al., 2024)). Therefore, we propose a statistical "CA spike detector" to locate these triggers on-the-fly, leading to dynamic (per-prompt) and surgical (intervening on required positions) cross-attention adjustment that improves on prior work.

## 4. Introducing the GUARD

GUARD is an inference-time memorization mitigation framework that, given a text prompt, modifies the image denoising process in a specialized way to guard against reproducing an original training image, while still producing a high-quality and prompt-aligned image.

GUARD achieves this via a modification of the standard CFG. Specifically, it steers the generation away from memorized content ("repulsion") and simultaneously, towards a high-quality but distinct-from-training-data alternative ("attraction"). The attraction towards a positive target serves two purposes: (i) to lower memorization, due to providing a safe target to redirect towards, and (ii) to maintain high quality; without an alternative high-quality target to redirect to, the use of the repulsion term can otherwise inadvertently harm quality while reducing memorization.

Let $\epsilon_\theta(x_t, e_\phi)$ denote the unconditional noise prediction at denoising timestep $t$, and let $\epsilon_\theta^-(x_t, e_p)$ denote the noise prediction conditioned on the original (memorized) prompt. Standard CFG computes the guided noise prediction as

$$\hat{\epsilon} = \epsilon_\theta(x_t, e_\phi) + s\left(\epsilon_\theta^-(x_t, e_p) - \epsilon_\theta(x_t, e_\phi)\right) \quad (4)$$

where $s$ is the guidance scale. This allows to control the degree of adherence to the given prompt, but may cause generating original training data. In contrast, GUARD redirects the generation away from the original prompt and towards a newly-added positive conditional noise prediction term $\epsilon_\theta^+(x_t, e_p)$, as follows:

$$\hat{\epsilon} = \epsilon_\theta(x_t, e_\phi) + s\left(\epsilon_\theta^+(x_t, e_p) - \epsilon_\theta(x_t, e_\phi)\right) \\ - r\left(\epsilon_\theta^-(x_t, e_p) - \epsilon_\theta(x_t, e_\phi)\right) \quad (5)$$

where $s$ controls attraction toward the "positive target" and $r$ controls repulsion from the "negative target", i.e. the (noise prediction for the) memorized prompt. Notably, GUARD's "positive target" is not an image target; it is a (prompt-)conditional noise prediction that aims to steer the denoising process away from a memorized image and towards an alternative high-quality image.

GUARD allows multiple instantiations through the choice of the positive target $\epsilon_\theta^+(x_t, e_p)$. We next present a concrete novel instantiation.

## 5. Instantiating the GUARD

Section 4 introduced GUARD as an abstract inference-time framework. However, GUARD can only be successful at memorization mitigation without quality degradation if an appropriate positive target is defined that GUARD will steer the generation towards. We therefore devote this section to the pursuit of such a positive target, i.e. an alternative denoising procedure that can produce a high-quality image that is distinct from training data. Towards that goal, we revisit a fundamental aspect of generating an image conditional on a text prompt, namely the cross-attention mechanism. We seek to understand the behaviour of the cross-attention distribution for different types of memorization, and accordingly develop a mechanism that adjusts that distribution to prevent memorization while maintaining quality. Based on this, this section contributes a GUARD by selecting a concrete positive target: a prompt-aligned conditioning signal obtained by attenuating cross-attention (CA) at prompt-specific memorization-critical locations.

### 5.1. You Don't Need All That (Cross-)Attention

Prior work has shown that the main driver of reproducing a training image is the memorization of its associated text prompt; a phenomenon that is tied with disproportionately large attention concentration on specific "trigger tokens" that the generation process can latch on to in order to retrieve the original image (Wen et al., 2024; Ren et al., 2024). To address this issue, we therefore seek to modify the attention distribution to prevent memorization triggers from receiving "all that attention". But which exactly are the token positions whose attention we should reduce? En route to discovering a fit-for-purpose attention redistribution mechanism, we first contribute an analysis of the attention patterns of (i) data points that are memorized versus data points that are not memorized, and (ii) for different types of memorization.

First, as seen in Figure 1a for verbatim memorization, the EOT token exhibits a large spike (a finding corroborated in (Ren et al., 2024) and leveraged in their mitigation method). In fact, Figure 1c shows that, throughout inference steps, the CA mass on the EOT token is much higher for a memorized

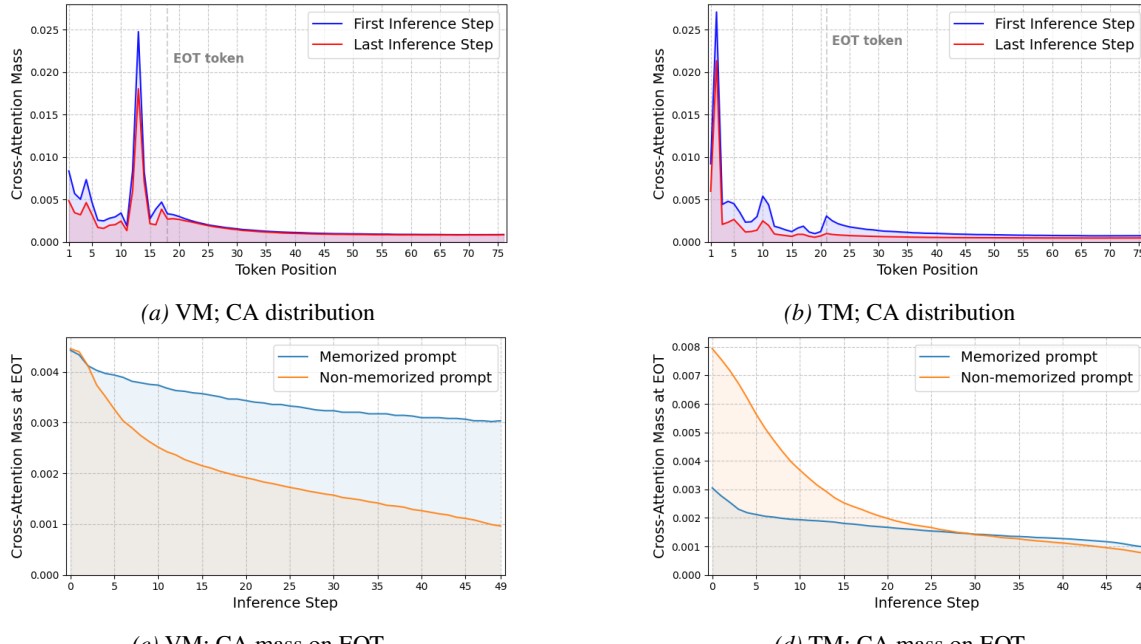

*(a)* VM; CA distribution

*(b)* TM; CA distribution

*(c)* VM; CA mass on EOT

*(d)* TM; CA mass on EOT

*Figure 1.* CA patterns. **(a) and (b)**: the CA mass across tokens, in the first and last inference steps for verbatim and template memorization (VM and TM). For clarity, we exclude the first token (position 0) from the plots, as it consistently receives the majority of CA across both memorized and non-memorized examples, which would dominate the scale and obscure differences among the remaining tokens. **(c) and (d)**: the CA mass on EOT across steps, for a memorized and a non-memorized prompt. **(c)**: VM on SD v1.4. **(d)**: TM on SD v2.0.

prompt than for a non-memorized one. However, note that Figure 1a shows that several other tokens also exhibit sharp spikes, often surpassing the magnitude of the EOT spike in fact. Thus, going beyond prior findings, we highlight that attenuating only at EOT, without special attention to these other spikes, is not a precise enough intervention.

Second, as seen in Figures 1b and 1d for template memorization, multiple tokens exhibit pronounced spikes, but unlike the verbatim case, the cross-attention assigned to EOT in typical memorized examples is no longer higher than that of non-memorized examples (comparing Figure 1d with Figure 1c). In fact, during the early inference steps, EOT cross-attention is *lower* for memorized examples, with the two converging at later steps. This reveals that attenuating CA at EOT, for example, is counter-productive in the case of template memorization, even if it worked well for verbatim memorization. This is corroborated by and explains our evaluation findings whereby the method in (Ren et al., 2024) fails to improve or even underperforms prior state-of-the-art methods for template memorization (Table 1, SD v2.0).

Put together, these findings suggest that there is no fixed rule that describes how cross-attention should be redistributed for memorization mitigation across examples and types of memorization. Therefore, a key desideratum for our attention redistribution strategy is to be dynamic and able to detect locations that require attenuation *per prompt*.

Overall, we seek a principled method that redistributes the (cross-)attention in a way that (i) disrupts the model's tendency to place substantial weight on "memorization trigger tokens", (ii) is "surgical" in that it maintains all other patterns that are needed for the generation to have high semantic alignment with the prompt, and (iii) is "dynamic" and adapts the reweighing strategy to the given prompt, since different prompts are associated with different attention patterns and necessitate different treatment. In the next subsection, we present a method that meets these desiderata.

### 5.2. Detecting and Attenuating Attention at Spikes

Our proposed cross-attention redistribution method has two parts: (i) an automatic detection of prompt-specific CA "spikes", and (ii) attenuation at those discovered spikes.

**Locating spikes.** We develop an attention-spike detector based on finding statistical outliers in the CA distribution, allowing to locate prompt-specific spikes on-the-fly. Concretely, we extract CA distribution maps conditioned under the embedding of the memorized prompt, $e_p$. We then score token-level spikes as follows: for each token position $i$, we compute the maximum attention mass assigned to that token across spatial queries (aggregating over specific blocks/heads as specified later):

$$M_i = \max_q \text{AttnScore}_{q,i}.$$

We compute the mean $\mu$ and standard deviation $\sigma$ over $\{M_i\}_i$ and define $Z_i = (M_i - \mu)/\sigma$. Tokens with $Z_i > \tau$ (e.g. $\tau = 3$) are flagged as "spiky". Let

$$\mathcal{S}(p) = \{\, i : Z_i > \tau \,\}$$

denote the set of spike positions for prompt $p$ (note that this set can include the EOT position).

**CA Attenuation at Spikes.** We next attenuate the detected CA spikes. Concretely, we scale down the corresponding CA logits by a multiplicative factor, thereby suppressing its influence on generation. This surgical attenuation yields a modified conditioning signal that remains aligned with the original prompt but with reduced memorization.

Given a memorized prompt $p$, in each cross-attention layer, we modify the attention logits prior to softmax by scaling down the logits corresponding to tokens in $\mathcal{S}(p)$:

$$\ell_{q,i} = \frac{\langle Q_q, K_i \rangle}{\sqrt{d}} \quad \longrightarrow \quad \ell'_{q,i} = \ell_{q,i} \cdot \alpha^{\mathbf{1}[i \in \mathcal{S}(p)]},$$

where $\alpha > 0$ controls attenuation strength.

We denote the modified noise prediction network obtained by the above attention redistribution mechanism, as $\epsilon_\theta^+(x_t, t, e_p; \mathcal{S}(p), \alpha)$, for a given prompt $p$, which now also depends on the set of automatically-discovered attention spikes $\mathcal{S}(p)$ and the scaling factor $\alpha$, treated as a hyperparameter. For a given prompt $p$, we instantiate GUARD by using $\epsilon_\theta^+(x_t, t, e_p; \mathcal{S}(p), \alpha)$ as the positive target. We refer to this instantiation of GUARD as CA-in-GUARD. We illustrate CA-in-GUARD in a diagram in Figure 2.

Note that by setting $r = 0$, CA-in-GUARD reduces to a novel method based solely on CA attenuation, which will be shown to outperform the prior state-of-the-art which leverages CA attenuation.

A diagram of the overall inference procedure of CA-in-GUARD is shown in Figure 2

### 5.3. Overall Inference-Time Procedure

For a given prompt $p$ and at each denoising step $t$, the simplest way to think of the proposed method is that it performs three U-Net evaluations: (1) an **unconditional** pass to obtain $\epsilon_\theta(x_t, t, e_\phi)$; (2) a **memorized conditional** pass (with standard CA) to obtain $\epsilon_\theta^-(x_t, t, e_p)$ and to extract CA maps for spike detection ; (3) a **spike-attenuated conditional** pass (same $e_p$, but with CA logits attenuated at $\mathcal{S}(p)$) to obtain $\epsilon_\theta^+(x_t, t, e_p; \mathcal{S}(p), \alpha)$. The final prediction $\hat{\epsilon}(x_t, t)$ is then formed by Eq. (5) and used to obtain $x_{t-1}$. In practice, we compute the three noise predictions in a single forward pass of the U-Net per timestep, by batching the null prompt, the positive neighbor conditioning, and the memorized prompt together, yielding efficiency close to that of a single forward pass; see Section A.2 for more details.

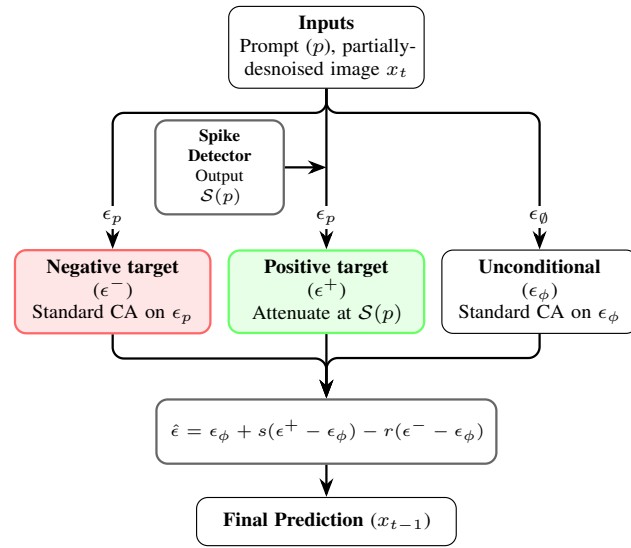

*Figure 2.* Overview of the CA-in-GUARD denoising process.

**How $\mathcal{S}(p)$ is computed.** We compute spike scores $Z_i$ from the per-prompt CA distributions extracted under the memorized conditional pass, using $M_i = \max_q \text{AttnScore}_{q,i}$ and thresholding $Z_i > \tau$, then set $\mathcal{S}(p) = \{i : Z_i > \tau\}$.

**When $\mathcal{S}(p)$ is computed.** We compute $\mathcal{S}(p)$ at every diffusion step and apply attenuation adaptively until no spikes are detected. This continuous strategy allows us to capture and suppress both early and late-stage attention spikes.

**Which blocks/heads are modified.** CA-logit attenuation is applied only in selected cross-attention modules of the U-Net. By default, we apply it in the *down* and *mid* blocks, , while omitting late *up* blocks to avoid quality degradation. Within each selected module, attenuation can be applied either to all heads, or restricted to *hot heads* (heads that allocate unusually large mass to $\mathcal{S}(p)$). We found that applying attenuation to all heads is a strong and simple default. Section A.5.1 in the Appendix discusses this in more detail.

## 6. Experimental Protocol

This section describes our evaluation protocol, outlining and justifying important differences over prior work. We depart from the protocol used in prior work in two important ways. First, we exclude examples that have relatively low memorization scores to begin with, as those examples are easy for all methods, so their inclusion may hide inherent problem difficulties and overestimate achievable performance. Second, we report results separately for verbatim vs template memorization, which illuminates previously-undiscovered issues in performance of certain methods.

**SD models and datasets.** We conduct experiments on the dataset of 500 memorized samples identified by Web-

ster (2023), which contains both verbatim and template memorization cases. Our experiments use Stable Diffusion (SD) v1.4 (pretrained on LAION-2B (Schuhmann et al., 2022)) and SD v2.0 (pretrained on a deduplicated subset of LAION-5B (Schuhmann et al., 2022)). Because verbatim memorization is much rarer in newer models pretrained on deduplicated data (Webster, 2023), we study both verbatim and template memorization on the earlier SD v1.4, but focus on template memorization only for SD v2.0.

**Metrics.** We quantify memorization using the SSCD score that measures the similarity between the image generated with a given prompt compared to the original training image that was associated with that prompt (Pizzi et al., 2022), where higher SSCD indicates stronger memorization. We measure generation quality using CLIP score (Radford et al., 2021) (higher indicates better prompt-image alignment) and FID (Heusel et al., 2017) (lower indicates better realism and diversity). We compute FID between images generated from the memorized prompts and a reference set of 10,000 images sampled from LAION (Schuhmann et al., 2022).

**Memorized prompt selection.** Unlike recent work (Han et al., 2025), we further filter memorized prompts and retain only highly memorized examples, i.e., having an SSCD score $> 0.7$. Our results reveal that omitting this filtering can hide the difficulties associated with memorization mitigation and provide a false sense of safety. Averaging SSCD scores over both highly and mildly memorized examples can artificially inflate the memorization mitigation performance of algorithms, which otherwise would struggle. This is clearly evident in template memorization and SD v2.0, where many previously identified memorized examples exhibit low SSCD scores. Hence, it is unsurprising that Han et al. (2025) (and the other baselines) reports substantially better SSCD performance (lower SSCD values), as their evaluation starts from a lower memorization baseline.

**Baselines.** We evaluate our method against prior state-of-the-art baselines, including Random Token Addition (RTA) (Somepalli et al., 2023b), Wen et al. (2024), Ren et al. (2024), Han et al. (2025), and the original model without any memorization mitigation as the reference point, to assess their ability to mitigate the memorization via an inference-time intervention.

**Hyperparameter selection.** There is a multitude of diverse criteria and trade-offs that we must take into consideration. We therefore adopt a principled selection protocol to ensure a fair and comprehensive comparison. We first run SD v1.4 and v2.0 on a subset of their pretraining data to obtain reference CLIP scores that represent strong image generation quality, which we aim to preserve. The rationale is that although some methods may substantially reduce memorization, this is of no practical use if image quality degrades severely. Said reference points were found to be

0.299 for SD v1.4 and 0.323 for SD v2.0. We allow up to a 15% degradation from these references, providing flexibility to identify configurations that trade moderate quality loss for strong memorization mitigation. Among runs within this acceptable range, we report the best results for each method in terms of SSCD, CLIP, and FID. For SD v1.4 with verbatim memorization, this yields a minimum acceptable CLIP score of 0.254. For template memorization, where all methods (including no mitigation) are far below the pre-training reference, we instead use the no-mitigation CLIP score as the reference and again allow a 15% degradation, yielding minimum acceptable CLIP scores of 0.186 for SD v1.4 and 0.183 for SD v2.0. Throughout the next section, "best CLIP" refers to the highest CLIP score achieved within these acceptable ranges.

For completeness, we also report results obtained using the traditional evaluation protocol of prior baselines, which does not apply this hyperparameter selection strategy. These results are shown in Figure 13 in Section A.7.2.

# 7. Experimental Results

Memorization mitigation must satisfy two criteria. First, achieve strong results in terms of any given metric, allowing a developer to configure it to achieve best results for their prioritized metric, for their specific use case. Second, it must incur minimal degradation in other metrics of interest when configured to work as well as possible for a given metric. We thus analyze the performance of methods in two ways: (i) we investigate their best achievable SSCD, CLIP, and FID, where the "best achievable" value for a metric is obtained by selecting the best configuration for that metric individually (in Figure 3), and then (ii) we explore what is the effect on SSCD, a primary metric, when configuring for achieving the best CLIP (in Figure 4, and in full detail in the Appendix Table 5) and what is the effect on CLIP and FID metrics metrics when selecting the best configuration for SSCD (in Table 1 and in full detail in Table 5), thus illuminating methods' trade-offs between different metrics. We describe our main findings below.

**Our CA attenuation alone outperforms prior work**. We begin by comparing our CA attenuation method against the previous state-of-the-art method of Ren et al. (2024), which is also based on CA attenuation and is thus is the most closely related prior method. Figure 3 and Table 1 show that our CA attenuation method alone significantly improves upon Ren et al. (2024) in terms of mitigating memorization, as shown by the SSCD metric, *across all settings* of architecture version and memorization type. The gap is particularly large especially for template memorization, and especially for SD v2.0. Concretely, our CA attenuation yields an SSCD of 0.53 (vs 0.60 for Ren et al. (2024)) for template memorization on SD v1.4, and 0.19 (vs 0.36 for

*Table 1.* Evaluation results across metrics, architectures and memorization types. For each setting (architecture and memorization type), we select for each method the configuration that yields the best SSCD, since this is the primary metric for memorization mitigation (in Figure 4, and in more detail in Table 5 in the Appendix, we show results with different selection criteria too). For each prompt, we generate four images and report the mean $\pm$(95% confidence interval) across four generations. Additional evaluation results on SD v3.0 are reported in Table 9 in A.8.

| Method | SD v1.4 – verbatim memorization | | | SD v1.4 – template memorization | | | SD v2.0 – template memorization | | |
|---|---|---|---|---|---|---|---|---|---|
| | SSCD ($\downarrow$) | CLIP ($\uparrow$) | FID ($\downarrow$) | SSCD ($\downarrow$) | CLIP ($\uparrow$) | FID ($\downarrow$) | SSCD ($\downarrow$) | CLIP ($\uparrow$) | FID ($\downarrow$) |
| No mitigation | 0.875 $\pm$0.001 | 0.346 $\pm$0.001 | 243.056 | 0.776 $\pm$0.017 | 0.219 $\pm$0.007 | 258.976 | 0.735 $\pm$0.011 | 0.215 $\pm$0.005 | 303.266 |
| RTA | 0.328 $\pm$0.007 | 0.263 $\pm$0.002 | 175.866 | 0.617 $\pm$0.043 | 0.187 $\pm$0.010 | 218.343 | 0.543 $\pm$0.048 | 0.183 $\pm$0.009 | 233.580 |
| Wen et al. | 0.115 $\pm$0.011 | 0.267 $\pm$0.003 | 162.848 | 0.545 $\pm$0.038 | 0.188 $\pm$0.008 | 209.719 | 0.260 $\pm$0.026 | 0.183 $\pm$0.008 | 188.914 |
| Ren et al. | 0.113 $\pm$0.007 | 0.258 $\pm$0.005 | 164.638 | 0.602 $\pm$0.033 | 0.184 $\pm$0.007 | 222.066 | 0.356 $\pm$0.024 | 0.188 $\pm$0.007 | 208.416 |
| Han et al. | 0.191 $\pm$0.016 | 0.256 $\pm$0.008 | 166.551 | 0.479 $\pm$0.033 | 0.188 $\pm$0.006 | 210.839 | 0.401 $\pm$0.024 | 0.186 $\pm$0.005 | 208.852 |
| CA attenuation | 0.109 $\pm$0.006 | 0.282 $\pm$0.004 | 164.660 | 0.530 $\pm$0.038 | 0.185 $\pm$0.009 | 212.240 | 0.193 $\pm$0.014 | 0.184 $\pm$0.005 | 245.850 |
| CA-in-GUARD | 0.079 $\pm$0.007 | 0.266 $\pm$0.015 | 158.115 | 0.517 $\pm$0.038 | 0.186 $\pm$0.008 | 210.983 | 0.193 $\pm$0.014 | 0.183 $\pm$0.005 | 212.727 |

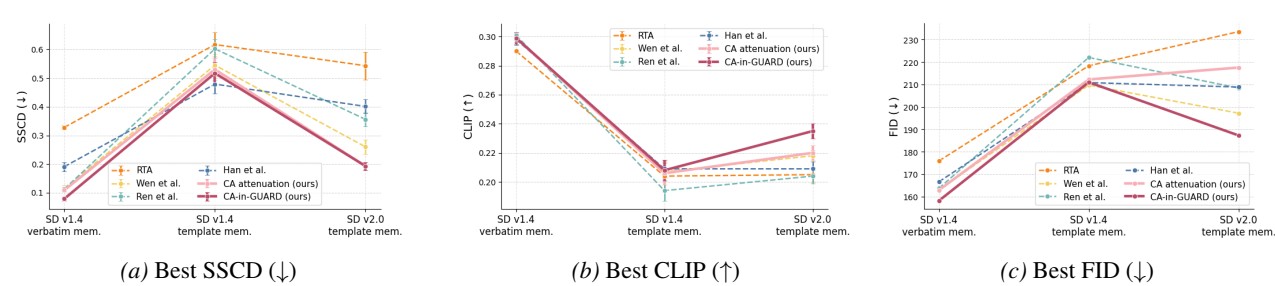

*(a)* Best SSCD ($\downarrow$)    *(b)* Best CLIP ($\uparrow$)    *(c)* Best FID ($\downarrow$)

*Figure 3.* **The *best achievable* SSCD, CLIP, and FID of different methods**. We plot the best value a method can achieve on each metric *individually*, using the configuration that yields best results on that specific metric. In each subplot, a different configuration may be used (we pick the hyperparameter setting that yields the best SSCD, best CLIP and best FID, respectively), so this plot does not speak to the ability of a method to do well on all metrics *jointly*, nor to trade-offs between these metrics, which we investigate separately.

Ren et al. (2024) for template memorization on SD v2.0, which are drastic gaps. This large improvement stems directly from our analysis on CA distribution and the findings (Figure 1) that attenuating CA only at EOT is insufficient.

The improved memorization mitigation of our CA attenuation over prior methods for CA attenuation, however, comes at some cost of quality on the FID metric (see Figure 3c), while CLIP is comparable. Figure 12 in Section A.7.1 shows that we can fully recover and boost quality significantly beyond prior work by incorporating this method into GUARD.

**CA-in-GUARD improves even more.** Having established that our CA attenuation alone improves upon the prior state-of-the-art for SSCD, our next finding is that we can gain further improvements by encasing our CA attenuation as the positive target of the GUARD framework, yielding the CA-in-GUARD method. Specifically, Figure 3 reveals that, compared to plain CA attenuation, CA-in-GUARD can further improve (or maintain equally good) performance for every single metric and setting. We note in particular an improvement in SSCD (SD v1.4, verbatim memorization) and large improvements in terms of CLIP and FID (especially in SD v2.0), mitigating the drawback of plain CA attenuation mentioned above. We hypothesize that allowing the repulsion term of GUARD to also play a part in memorization mitigation can alleviate the pressure of strict CA spike

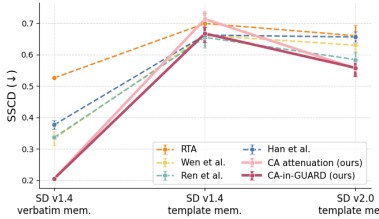

*Figure 4.* **Comparison of the SSCD of different methods *for the hyperparameter configuration that works best for CLIP*.**

reduction, which may come at a cost of quality, if acting as the sole memorization mitigator. This observation suggests that the repulsion term of GUARD acts synergistically with the spike attention attenuation mechanism to yield good trade-offs between mitigating memorization while preserving quality (Figure 12 in Section A.7.1).

**CA-in-GUARD dominates across settings**. A key overarching conclusion is that the state-of-the-art among all prior research is not a single method, but changes depending on the setting. For example, Table 1 shows that for SD v1.4 template memorization, Han et al. (2025) is the strongest performer for SSCD (0.479) out of prior work (excluding our methods). However, Han et al. (2025) is outperformed by Wen et al. (2024) and Ren et al. (2024) for template memorization on SD v2.0, for example. Generally, no prior

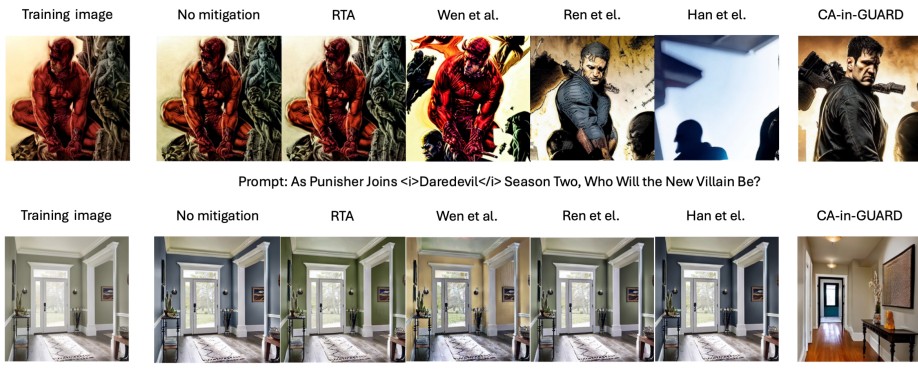

*Figure 5.* Qualitative examples on SD v1.4 under verbatim (top row) and template memorization (bottom row). No mitigation often reproduces the training example closely. CA-in-GUARD can significantly mitigate memorization while preserving prompt-relevant content and image quality. Additional examples are provided in A.9.

method performs robustly across settings. On the other hand, CA-in-GUARD provides overall better performance across all settings considered, both when looking at its per-metric best achievable results relative to the per-metric best achievable results of other methods (Figure 3), as well as its trade-offs across metrics: Figure 4 shows that CA-in-GUARD by far improves on the SSCD of prior methods, even when configuring for best CLIP, and Table 1 shows that, when all methods are configured for best SSCD, CA-in-GUARD outperforms prior methods on nearly all settings in terms of SSCD and sometimes in terms of CLIP and FID too, otherwise yielding competitive but not improved results on the quality metrics while improving SSCD. Additional evaluations further support this conclusion: CA-in-GUARD remains robust across samplers, step counts, guidance schedules, and CFG scales (Section A.6); remains top-performing under DINO-based retrieval metrics (Caron et al., 2021)(Sections A.7.3); and continues to achieve the strongest overall SSCD results over the full 500-example spectrum, rather than only the high-memorization subset (Section A.7.4). Qualitative examples in Figure 5 (and Section A.9) show the same trend visually, with CA-in-GUARD moving generations away from memorized training images while preserving prompt-relevant content. Moreover, we go beyond the standard prior-work setting on SD v1.4 and SD v2.0 by reporting an evaluation on SD v3.0 in Section A.8, which, to our knowledge, presents the first evaluation of any memorization mitigation methods on SD v3.0. CA-in-GUARD continues to outperform the applicable baselines on this newer architecture, suggesting that its empirical gains extend beyond the SD v1.4/v2.0 setting. Generally, CA-in-GUARD is by far the most consistent method in terms of performing strongly across metrics and settings.

**Mitigating template memorization is harder.** Across all baselines, template memorization consistently proves more challenging to mitigate than verbatim memorization. While the "no mitigation" reference point exhibits lower SSCD under template memorization (which indicates a weaker initial memorization signal), all baseline methods degrade substantially when moving from verbatim to template memorization, with worse SSCD, CLIP, and FID scores. This degradation persists in both SD v1.4 and SD v2.0, suggesting that template memorization is a more structurally challenging problem for existing mitigation approaches. Figure 3, 4 and Table 1 show that, across settings, CA-in-GUARD consistently outperforms prior methods on template memorization, yielding a large reduction in SSCD in many cases while having comparable CLIP and FID.

**Other analyses.** We also explore the efficiency of our method (Section A.7.5), other GUARD instantiations (Section A.4), and which U-Net blocks, heads, and denoising steps to operate on (Sections A.5.1, A.5.2, A.5.3, respectively).

# 8. Conclusion

We have proposed GUARD, a new framework for inference-time memorization mitigation in T2I diffusion models, and a concrete instantiation of it, CA-in-GUARD, based on an improved CA attenuation method that we developed, motivated by our empirical analyses. Our CA attenuation approach alone improves upon prior state-of-the-art in terms of memorization mitigation, and its integration with GUARD yields by far the most robust method in terms of consistently producing strong memorization mitigation results, across architectures (SD v1.4, v2.0, and v3.0) and memorization types (verbatim and template memorization), while also improving or remaining competitive with prior methods on image quality. We hope future work develops even better instantiations of GUARD, and investigates incorporating the fundamentals of our proposal into training-time or finetuning-time mitigation approaches.

## Acknowledgements

We thank Jamie Hayes for useful feedback on an earlier draft of this work.

## Impact Statement

This work explores the problem of memorization mitigation in text-to-image (T2I) diffusion models, with possible implications for improving privacy, copyright compliance, and responsible model deployment. As T2I systems become increasingly integrated into real-world applications, the ability to suppress the memorization of specific training examples is critical for addressing ethical, legal, and societal concerns.

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

# A. Appendix

## Contents

## A.1. Limitations

A limitation of our method, and indeed any inference-time memorization mitigation approach, is that it does not (even attempt to) erase memorized information from the model weights. It simply attempts to prevent it from overly affecting the generation process. While we operate under this framework by choice, to avoid other issues inherent in training-time and finetuning-time mitigation approaches, we acknowledge that the family of inference-time methods has the inherent limitation of not being able to protect against any threat model (e.g. an adversary that has white-box access to the weights can still retrieve the memorized information). Nonetheless, we believe that inference-only mitigation methods can be practically very important, under realistic black-box threat models, and due to their promise to yield better trade-offs in terms of efficiency, memorization mitigation and quality preservation. Within the inference-time mitigation category, our method outperforms all prior

work. Given this, an important area for future work is to integrate insights from our method into training-time and / or finetuning-time mitigation methods to investigate whether we can push the pareto frontier further for those categories too.

## A.2. Implementation Details

**Evaluation configuration and hardware.** Following prior work on memorization mitigation (Han et al., 2025; Ren et al., 2024; Wen et al., 2024), we evaluate our methods on two Stable Diffusion (SD) models: SD v1.4 and SD v2.0. Since SD v2.0 was pretrained on a de-duplicated dataset, verbatim memorization is substantially reduced compared to SD v1.4; however, it still exhibits template memorization, as noted by Webster (2023). We therefore evaluate memorization under three settings: (i) SD v1.4 with verbatim memorization, (ii) SD v1.4 with template memorization, and (iii) SD v2.0 with template memorization.

We start from the 500 memorized prompts from the LAION dataset identified by Webster (2023). For each setting, we generate images using the corresponding model architecture and compute SSCD scores for all prompts. To focus on genuinely memorized cases, we retain only prompts with $SSCD > 0.7$. This filtering yields 72 prompts for SD v1.4 with verbatim memorization, 143 prompts for SD v1.4 with template memorization, and 96 prompts for SD v2.0 with template memorization. For all experiments, we generate 4 images per prompt using 50 inference steps. Results are reported as mean values with 95% confidence intervals for SSCD and CLIP. FID is computed over the entire generation set and therefore does not include confidence intervals.

All experiments were performed on NVIDIA A5000 GPUs, with a total computational cost of approximately 1,500 GPU hours.

**Batched forward pass.** We concatenate the unconditional, memorized conditional, and spike-attenuated conditional inputs into a single batch, and then enable the U-Net to process all streams simultaneously in one forward pass. Within each cross-attention layer, we intercept the intermediate attention logit tensor and dynamically slice it along the batch dimension to isolate the spike-attenuated conditional stream. We then compute token-wise statistics (i.e., Z-scores) on this specific slice to detect outlier "spikes" and apply an in-place attenuation to the logits immediately before the softmax operation. Thus, we effectively suppress CA of memorization triggers in real-time without disrupting the parallel computation of other streams. A diagram of the overall inference procedure of CA-in-GUARD is shown in Figure 2.

**Hyperparameter Tuning.** We tune hyperparameters separately for each architecture-memorization setting and each

*Table 2.* Hyperparameter search ranges for CA attenuation and CA-in-GUARD across different settings.

| | $\tau$ | $\alpha$ | $r$ |
|---|---|---|---|
| SD v1.4 – verbatim mem. | [0.01, 0.05] | [0.05, 0.5] | / |
| SD v1.4 – template mem. | [0.5, 3.0] | [0.1, 0.7] | / |
| SD v2.0 – template mem. | [0.5, 3.0] | [0.1, 0.7] | / |

*(a)* CA attenuation

| | $\tau$ | $\alpha$ | $r$ |
|---|---|---|---|
| SD v1.4 – verbatim mem. | [0.1, 0.3] | [0.1, 0.5] | [0.1, 2.0] |
| SD v1.4 – template mem. | [0.5, 1.5] | [0.5, 0.9] | [0.1, 2.0] |
| SD v2.0 – template mem. | [1.5, 3.0] | [0.5, 0.9] | [2.0, 6.0] |

*(b)* CA-in-GUARD

mitigation method. For CA attenuation, we consider two hyperparameters: the scaling factor $\alpha$, which controls the strength of CA attenuation, and the threshold $\tau$ used for spike detection (see Section 5.2). For CA-in-GUARD, in addition to $\alpha$ and $\tau$, we introduce an extra hyperparameter $r$, which controls the strength of the negative target term in the GUARD framework (see Eq. 5 in Section 4). We perform grid search over method-specific hyperparameter ranges for each setting. The ranges explored for CA attenuation and CA-in-GUARD are summarized in Table 2.

**Code.** The code for reproducing the results is available at: `https://github.com/kairanzhao/GUARD`

### A.3. Robustness of CA Attenuation on Non-Memorized Prompts

An important practical question is how memorization mitigation affects prompts that are not memorized by the model. While prior work typically assumes that memorized examples are identified beforehand and mitigation is applied selectively at inference time, we investigate whether CA attenuation negatively impacts generation quality on non-memorized inputs.

We evaluate this using a subset of 200 prompts from the LAION dataset (Schuhmann et al., 2022), which was used during pretraining of both SD v1.4 and v2.0. These prompts serve as our non-memorized examples. As a reference, we first generate images using the pretrained models without any mitigation. The resulting SSCD scores are very low for both models (0.074 for SD v1.4 and 0.069 for SD v2.0, as shown in Table 3), confirming that these prompts are indeed not memorized.

We then apply our CA attenuation method using the same hyperparameters that achieve strong mitigation under the memorization settings, without any tuning for this non-memorized regime. We evaluate performance on the same set of prompts for both SD v1.4 and SD v2.0. The results

are reported in Table 3.

*Table 3.* Comparison of no mitigation and CA attenuation on non-memorized prompts for Stable Diffusion v1.4 and v2.0. Results are reported as mean values with 95% confidence intervals.

| Method | SD v1.4 | | | SD v2.0 | | |
|---|---|---|---|---|---|---|
| | SSCD | CLIP | FID | SSCD | CLIP | FID |
| No mitigation | $0.071_{\pm 0.006}$ | $0.299_{\pm 0.010}$ | 141.947 | $0.074_{\pm 0.006}$ | $0.322_{\pm 0.006}$ | 141.350 |
| CA attenuation | $0.069_{\pm 0.006}$ | $0.298_{\pm 0.006}$ | 142.898 | $0.072_{\pm 0.006}$ | $0.320_{\pm 0.005}$ | 139.417 |

Across all metrics, we observe no statistically significant difference between the no-mitigation baseline and CA attenuation on the non-memorized dataset. These promising findings suggest that CA attenuation is robust to non-memorized inputs and does not degrade generation quality when memorization is absent. *Importantly, this relaxes a common assumption made by prior work: mitigation does not require (methods to accumulate) prior knowledge of which prompts are memorized.* Instead, CA attenuation can be applied universally at inference time, providing protection against memorization without harming non-memorized generations. A systematic evaluation of the potential negative impact of other memorization mitigation methods on non-memorized examples is equally important, but we leave such a comprehensive evaluation to future work.

### A.4. Alternative Attract Components in GUARD

We next analyze the role of the *attract* component in GUARD through ablations, by using alternative definitions of "positive targets" to attract toward. We focus on a comparison between two classes of positive targets: (i) CA-attenuation-based targets, i.e., attenuating cross-attention at memorization-relevant token locations, and (ii) semantic targets, i.e., using a paraphrased sequence of the original memorized prompt and use its predicted noise as the target. Figure 6 compares these two variants across all memorization settings.

As shown in Figure 6, CA-attenuation-based targets consistently outperform semantic targets as the positive target choice for GUARD across all three settings. This suggests that directly manipulating cross-attention dynamics provides more effective memorization mitigation while preserving generation quality, compared to semantic paraphrasing. In the following, we further ablate different strategies for applying CA attenuation within the GUARD framework.

### A.5. Ablations for CA Attenuation

#### A.5.1. U-Net Block Identification and Ablation

Section 5.1 shows that memorization is associated with abnormally high CA values that persist throughout the generation process, consistent with prior findings (Ren et al., 2024). Attenuating CA at critical locations can therefore mitigate memorization. However, it remains unclear whether such

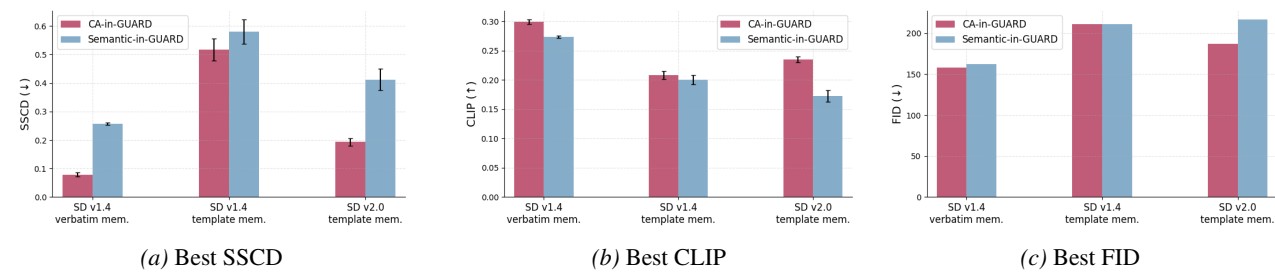

*(a)* Best SSCD        *(b)* Best CLIP        *(c)* Best FID

*Figure 6.* The *best achievable* SSCD, CLIP, and FID of CA-in-GUARD (our default) versus semantic-in-GUARD (ablation), evaluated across 3 settings: SD v1.4 with verbatim memorization, template memorization, and SD v2.0 with template memorization.

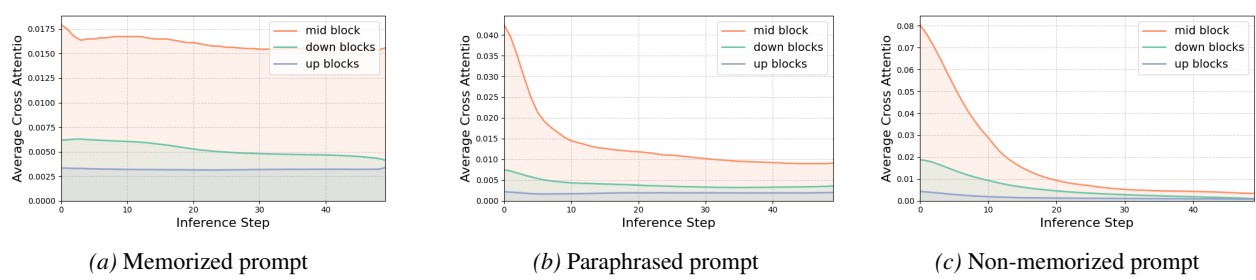

*(a)* Memorized prompt        *(b)* Paraphrased prompt        *(c)* Non-memorized prompt

*Figure 7.* Cross-attention mass in different U-Net blocks over inference steps, by comparing three type of prompts: (i) a memorized prompt, (ii) a counterfactual non-memorized prompt obtained by paraphrasing the memorized prompt (which already substantially reduces SSCD), and (iii) an unrelated non-memorized prompt

attenuation should be applied uniformly across all U-Net blocks. In particular, a more surgical strategy–selectively attenuating CA in blocks that are primarily responsible for memorization while preserving others that may be more important for generation quality–may achieve stronger mitigation with less degradation in output quality.

To study this, we analyze where memorization-related CA concentrates across the U-Net. We use the EOT token in Stable Diffusion v1.4 under verbatim memorization as a representative example. For each U-Net block, we track the average CA mass assigned to EOT over inference steps. We compare three prompts: (i) a memorized prompt, (ii) a counterfactual non-memorized prompt obtained by paraphrasing the memorized prompt (which already substantially reduces SSCD), and (iii) an unrelated non-memorized prompt.

The results in Figure 7 reveal a clear structural distinction. For the memorized prompt, CA at EOT is highest in the mid-block, followed by the down-blocks, and remains elevated throughout the generation process. In contrast, for both non-memorized prompts (Figures 7b and 7c), CA in the mid- and down-blocks, although sometimes high at early steps, decays rapidly as generation proceeds. CA in the remaining blocks is consistently low for both memorized and non-memorized prompts, indicating limited involvement in memorization dynamics. We therefore do not intervene on these blocks.

Motivated by this observation, our intervention selectively attenuates CA only in the mid- and down-blocks, aim-

ing to simulate the natural decay pattern observed in non-memorized generations by gradually reducing CA at memorization-trigger tokens. Figure 8 presents an ablation comparing block-selective attenuation with uniform attenuation across all blocks. Selective attenuation consistently yields a more favorable SSCD-CLIP-FID trade-off, as reflected by improved Pareto frontiers.

### A.5.2. ATTENTION HEADS IDENTIFICATION AND ABLATION

Following the identification of the most memorization-critical U-Net blocks (mid- and down-blocks) and the corresponding block-level ablations, we next investigate which attention heads within these blocks should be targeted. The goal is to further localize the intervention by focusing on the heads most responsible for memorization, while leaving the remaining heads unaffected.

For each attention head in the selected mid- and down-blocks, we compute a *hotness* score that captures how strongly and selectively the head attends to the memorization trigger token (e.g., the EOT token). Specifically, the score is defined as $H_i = \text{AttnScore}_i \times (1/\text{Entropy})$, where $\text{AttnScore}_i$ denotes the total attention mass assigned to the trigger token $i$, and Entropy denotes the dispersion of the head's attention distribution. This score prioritizes heads that concentrate attention sharply on the trigger token rather than distributing it diffusely across the prompt. We then, for each block, select the top $k$ heads with the highest hotness

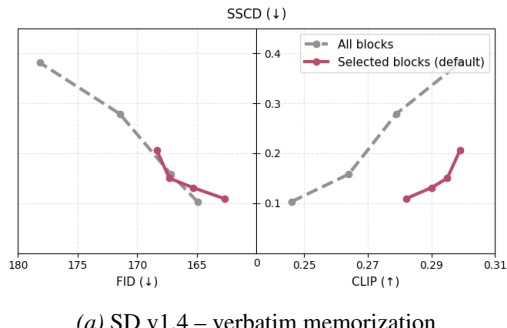
*(a)* SD v1.4 – verbatim memorization

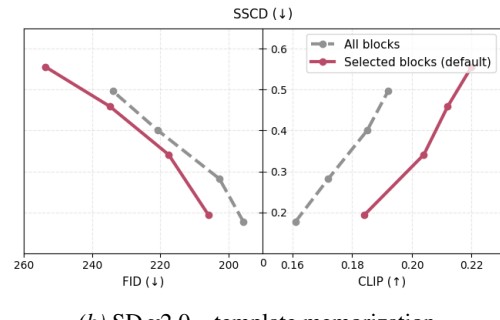
*(b)* SD v2.0 – template memorization

*Figure 8.* **SSCD–FID and SSCD–CLIP Pareto frontiers comparing CA attenuation using *selective blocks*(our default) vs. *all blocks*(ablation).** Lower SSCD and FID and higher CLIP indicate better performance; methods closer to the bottom-right corner are optimal in both the left and right subfigures.

scores and apply CA attenuation only to this subset of "hot" heads for the remainder of the generation process. We experiment with different proportions $k$ of selected heads per block to study the trade-off between memorization mitigation and generation quality.

The results in Figure 9. reveal a clear difference between memorization types. Under verbatim memorization, restricting CA attenuation to only a subset of heads substantially limits effectiveness: the lowest SSCD achievable is approximately 0.3, compared to around 0.1 when all heads are attenuated. In contrast, under template memorization, performance is much less sensitive to the number of selected heads. These observations indicate that selectively targeting heads is insufficient for robust mitigation in the verbatim memorization setting, and that attenuating all heads within the identified blocks is a stronger and more reliable default.

### A.5.3. TIMESTEPS IDENTIFICATION AND ABLATION

We next examine how the effectiveness of CA attenuation varies depending on when it is applied during the diffusion inference process. Specifically, we consider four strategies: (i) applying CA attenuation at all timesteps (our default); (ii) applying CA attenuation only during the first 50% of timesteps; (iii) applying CA attenuation at all timesteps with a cosine decay schedule; and (iv) applying CA attenuation at all timesteps with a linear decay schedule.

Figure 10 shows the corresponding SSCD–CLIP and SSCD–FID Pareto frontiers. Overall, applying CA attenuation at all timesteps emerges as a strong and robust default, as it consistently yields more favorable SSCD-CLIP trade-offs compared to other strategies.

### A.6. Ablations for CA-in-GUARD

For comparability with prior work, our main experiments use the standard inference configuration: DDIM sampling with 50 denoising steps and classifier-free guidance (CFG)

scale $s = 7.5$. To evaluate whether CA-in-GUARD depends on this specific setup, we conduct additional ablations over common Stable Diffusion inference choices, including samplers, denoising step counts, guidance schedules, and CFG scales. Results are shown in Figure 11. Overall, CA-in-GUARD remains robust and performs well across these settings.

**Sampler.** We compare DDIM, Euler A, and DPM++ samplers. CA-in-GUARD consistently reduces memorization across these samplers, indicating that its effectiveness is not tied to a particular sampling algorithm.

**Number of denoising steps.** We evaluate 30, 50, and 70 denoising steps. CA-in-GUARD remains effective under both shorter and longer sampling budgets, suggesting that the method is stable across different inference costs.

**Guidance schedule.** We compare three guidance schedules: no schedule, cosine decay, and linear decay. CA-in-GUARD performs reliably across these schedules, showing that the method does not require a finely tuned guidance trajectory.

**CFG scale.** We further ablate CFG scales $s \in \{1, 4, 7, 7.5\}$, where $s = 7.5$ is the default setting. Without mitigation, reducing $s$ lowers memorization but also degrades generation quality. This behavior is expected from the standard CFG update,

$$\epsilon_{\text{pred}} = \epsilon_{\text{uncond}} + s(\epsilon_{\text{cond}} - \epsilon_{\text{uncond}}),$$

where $s$ controls the strength of prompt-conditioned steering. Smaller values of $s$ weaken prompt alignment, which can reduce memorization but also harms fidelity.

CA-in-GUARD remains effective across CFG scales, consistently reducing SSCD relative to the no-mitigation baseline while maintaining a favorable quality trade-off. Its mitigation effect is weaker at $s = 1$, but this behavior is not specific to CA-in-GUARD: other inference-time mitigation methods also show limited improvement over the "no mitigation" baseline (Table 4). We interpret this as a consequence

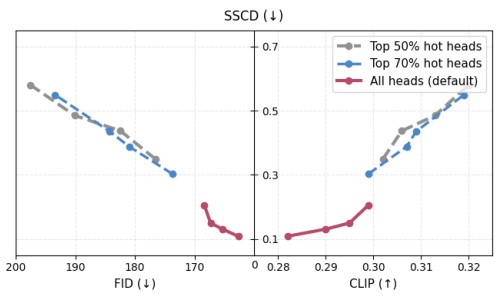
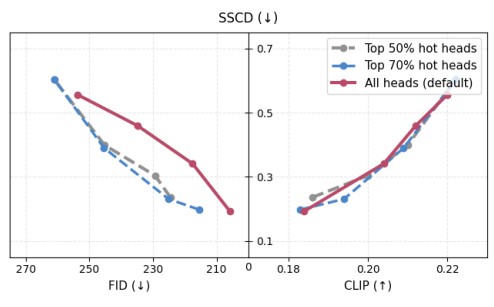

*(a)* SD v1.4 – verbatim memorization

*(b)* SD v2.0 – template memorization

*Figure 9.* **SSCD-FID and SSCD-CLIP Pareto frontiers comparing CA attenuation over *all heads* (our default) vs. *top k% heads* (ablation)**. Lower SSCD and FID and higher CLIP indicate better performance; thus, methods closer to the bottom-right corner are optimal for both left and right subfigures.

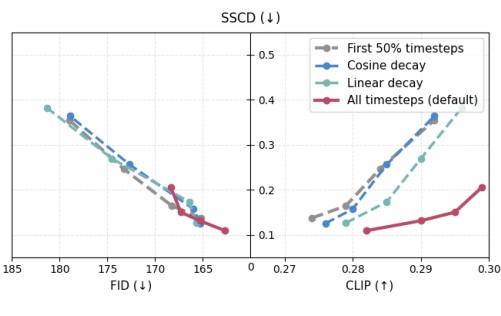
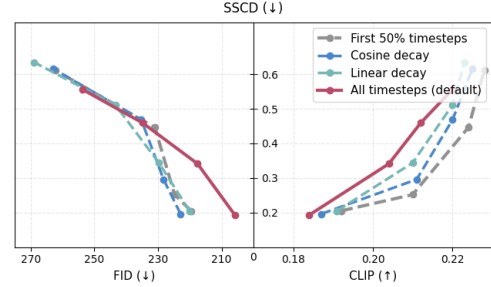

*(a)* SD v1.4 – verbatim memorization

*(b)* SD v2.0 – template memorization

*Figure 10.* **SSCD-FID and SSCD-CLIP Pareto frontiers comparing CA attenuation over *all timesteps* (our default) vs. *other variants* (ablation)**. Lower SSCD and FID and higher CLIP indicate better performance; thus, methods closer to the bottom-right corner are optimal for both left and right subfigures.

of the reduced prompt-conditioned guidance signal. Larger CFG scales amplify the prompt-conditioned direction, increasing both memorization risk and the structured signal that inference-time mitigation methods can intervene on. At $s = 1$, CFG reduces to plain conditional denoising, leaving less headroom for any inference-time method to operate on (because most inference-time methods operate by modifying the guidance signal or its induced trajectory). We leave a more systematic study of this interaction as future work.

*Table 4.* Quantitative results for the CFG scale ablation in Figure 11d, reported at CFG scale $s = 1.0$, with each method shown under the configuration with the best SSCD (our primary memorization-mitigation metric).

| Method | SSCD | CLIP | FID |
|---|---|---|---|
| No mitigation | $0.571 \pm 0.068$ | $0.280 \pm 0.019$ | 194.697 |
| Wen et al. | $0.569 \pm 0.064$ | $0.284 \pm 0.016$ | 188.374 |
| CA-in-GUARD | $0.403 \pm 0.071$ | $0.243 \pm 0.019$ | 184.982 |

### A.7. Detailed Results

#### A.7.1. CA-IN-GUARD VS. CA ATTENUATION VS. PRIOR WORK

This section presents Figure 12, a focused subset of the results from Figure 3, restricted to the methods most relevant for addressing the two specific questions in Section 7: (1) whether CA attenuation alone improves upon prior work, and (2) whether further gains can be achieved by incorporating CA attenuation into the GUARD framework (CA-in-GUARD).

To address the first question, we compare CA attenuation with Ren et al. (2024), which is the closest prior approach in spirit. As shown in Figure 12, CA attenuation consistently matches or outperforms Ren et al. (2024) across all architecture versions and memorization settings. The improvement is particularly pronounced in memorization mitigation (Figure 12a), where CA attenuation yields substantially lower SSCD under template memorization for both SD v1.4 and SD v2.0.

We then compare CA attenuation with CA-in-GUARD. Note that CA attenuation corresponds to a special case of CA-in-GUARD obtained by setting $r = 0$ in Eq. 5 (i.e., removing

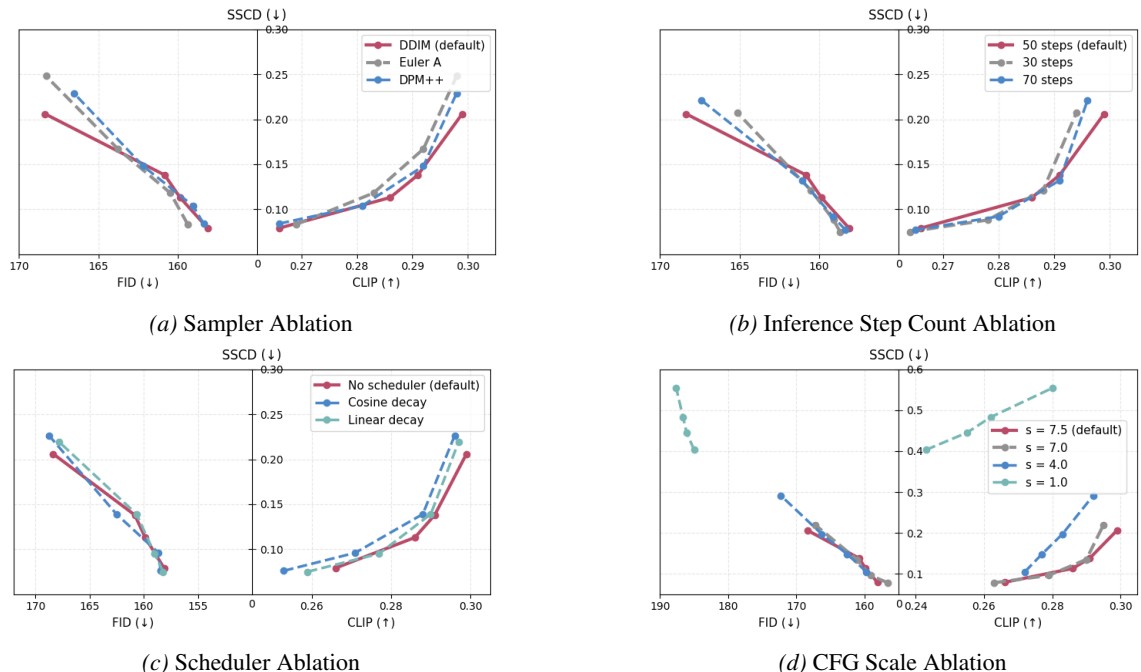

*Figure 11.* SSCD-FID and SSCD-CLIP Pareto frontiers for CA-in-GUARD ablations across *sampler choice*, *inference steps*, *scheduler choice*, and *CFG scale*. Lower SSCD and FID, together with higher CLIP, indicate better performance; methods closer to the bottom-right corner achieve a more favorable trade-off.

the negative target from GUARD). As shown in Figure 12, CA-in-GUARD consistently improves upon or maintains the performance of CA attenuation across all metrics and settings, with particularly clear gains in CLIP and FID under SD v2.0.

### A.7.2. EVALUATION VIA SSCD-CLIP AND SSCD-FID PARETO FRONTIERS

We present here the full results Table 5 obtained under our experimental protocol described in Section 6. This table provides detailed numerical results and serves as the basis for the summarized comparisons shown in Figures 3, 4 and 12.

In addition to our protocol-driven evaluation, we also report results using the traditional Pareto-front analysis commonly adopted in prior work (Wen et al., 2024; Ren et al., 2024; Han et al., 2025). Specifically, we construct SSCD-CLIP and SSCD-FID Pareto frontiers for each model version and memorization type setting. These results are shown in Figure 13.

Across all experimental settings, our methods achieve the best or comparable SSCD-CLIP and SSCD-FID trade-offs relative to existing baselines. The advantage is particularly notable for the SSCD-CLIP trade-off, where both CA attenuation and CA-in-GUARD consistently dominate other methods. The only exception occurs in the SD v1.4 tem-

plate memorization setting for SSCD–CLIP, where Han et al. (2025) attains a slightly better trade-off but performs significantly worse in most other settings. Overall, these results confirm that our methods remain competitive–and often superior–under traditional evaluation protocols, in addition to the more controlled evaluation protocol adopted in Section 6.

### A.7.3. EVALUATION UNDER SSCD AND DINO METRICS

To further validate the effectiveness of our methods under complementary perceptual similarity metrics, we evaluate template memorization using both SSCD and DINO-based retrieval scores. SSCD is widely used to measure perceptual similarity for verbatim and near-verbatim memorization, while DINO retrieval provides an additional feature-space measure of similarity to training-set neighborhoods (Caron et al., 2021) . Specifically, we report DINO@1 and DINO@5, corresponding to the retrieval similarity to the nearest and top-5 nearest training examples, respectively. Lower scores indicate weaker similarity to the training set and therefore less memorization.

As shown in Table 6, the conclusions are consistent across both SSCD and DINO metrics. CA attenuation substantially reduces memorization compared with the no-mitigation baseline and remains competitive with prior mitigation methods. CA-in-GUARD further improves or matches this per-

*Table 5.* Comparison of memorization mitigation methods across SD v1.4 and SD v2.0 under verbatim and template memorization settings. Results are evaluated using SSCD, CLIP, and FID. Each subtable shows results when selecting the best configuration for one target metric (SSCD, CLIP, or FID). For each prompt, we generate four images and report the mean $\pm$ (95% confidence interval) across the four generations.

| Method | SD v1.4 – verbatim memorization | | | SD v1.4 – template memorization | | | SD v2.0 – template memorization | | |
|---|---|---|---|---|---|---|---|---|---|
| | SSCD ($\downarrow$) | CLIP ($\uparrow$) | FID ($\downarrow$) | SSCD ($\downarrow$) | CLIP ($\uparrow$) | FID ($\downarrow$) | SSCD ($\downarrow$) | CLIP ($\uparrow$) | FID ($\downarrow$) |
| No mitigation | $0.875_{\pm 0.001}$ | $0.346_{\pm 0.001}$ | 243.056 | $0.776_{\pm 0.017}$ | $0.219_{\pm 0.007}$ | 258.976 | $0.735_{\pm 0.011}$ | $0.215_{\pm 0.005}$ | 303.266 |
| RTA | $0.328_{\pm 0.007}$ | $0.263_{\pm 0.002}$ | 175.866 | $0.617_{\pm 0.043}$ | $0.187_{\pm 0.010}$ | 218.343 | $0.543_{\pm 0.048}$ | $0.183_{\pm 0.009}$ | 233.580 |
| Wen et al. | $0.115_{\pm 0.011}$ | $0.267_{\pm 0.003}$ | 162.848 | $0.545_{\pm 0.038}$ | $0.188_{\pm 0.008}$ | 209.719 | $0.260_{\pm 0.026}$ | $0.183_{\pm 0.008}$ | 188.914 |
| Ren et al. | $0.113_{\pm 0.007}$ | $0.258_{\pm 0.005}$ | 164.638 | $0.602_{\pm 0.033}$ | $0.184_{\pm 0.007}$ | 222.066 | $0.356_{\pm 0.024}$ | $0.188_{\pm 0.007}$ | 208.416 |
| Han et al. | $0.191_{\pm 0.016}$ | $0.256_{\pm 0.008}$ | 166.551 | $0.479_{\pm 0.033}$ | $0.188_{\pm 0.006}$ | 210.839 | $0.401_{\pm 0.024}$ | $0.186_{\pm 0.005}$ | 208.852 |
| CA attenuation | $0.109_{\pm 0.006}$ | $0.282_{\pm 0.004}$ | 164.660 | $0.530_{\pm 0.038}$ | $0.185_{\pm 0.009}$ | 212.240 | $0.193_{\pm 0.014}$ | $0.184_{\pm 0.005}$ | 245.850 |
| CA-in-GUARD | $0.079_{\pm 0.007}$ | $0.266_{\pm 0.015}$ | 158.115 | $0.517_{\pm 0.038}$ | $0.186_{\pm 0.008}$ | 210.983 | $0.193_{\pm 0.014}$ | $0.183_{\pm 0.005}$ | 212.727 |

*(a)* Best SSCD

| Method | SD v1.4 – verbatim memorization | | | SD v1.4 – template memorization | | | SD v2.0 – template memorization | | |
|---|---|---|---|---|---|---|---|---|---|
| | SSCD ($\downarrow$) | CLIP ($\uparrow$) | FID ($\downarrow$) | SSCD ($\downarrow$) | CLIP ($\uparrow$) | FID ($\downarrow$) | SSCD ($\downarrow$) | CLIP ($\uparrow$) | FID ($\downarrow$) |
| No mitigation | $0.875_{\pm 0.001}$ | $0.346_{\pm 0.001}$ | 243.056 | $0.776_{\pm 0.017}$ | $0.219_{\pm 0.007}$ | 258.976 | $0.735_{\pm 0.011}$ | $0.215_{\pm 0.005}$ | 303.266 |
| RTA | $0.526_{\pm 0.003}$ | $0.290_{\pm 0.001}$ | 196.322 | $0.699_{\pm 0.033}$ | $0.204_{\pm 0.009}$ | 237.061 | $0.660_{\pm 0.034}$ | $0.205_{\pm 0.005}$ | 269.205 |
| Wen et al. | $0.333_{\pm 0.022}$ | $0.299_{\pm 0.003}$ | 174.106 | $0.661_{\pm 0.032}$ | $0.207_{\pm 0.007}$ | 230.140 | $0.630_{\pm 0.024}$ | $0.218_{\pm 0.004}$ | 263.483 |
| Ren et al. | $0.337_{\pm 0.007}$ | $0.301_{\pm 0.002}$ | 172.417 | $0.655_{\pm 0.033}$ | $0.194_{\pm 0.007}$ | 232.503 | $0.584_{\pm 0.024}$ | $0.204_{\pm 0.005}$ | 254.504 |
| Han et al. | $0.377_{\pm 0.014}$ | $0.298_{\pm 0.004}$ | 182.813 | $0.662_{\pm 0.024}$ | $0.209_{\pm 0.006}$ | 235.635 | $0.656_{\pm 0.018}$ | $0.209_{\pm 0.005}$ | 273.532 |
| CA attenuation | $0.206_{\pm 0.004}$ | $0.299_{\pm 0.004}$ | 168.356 | $0.679_{\pm 0.025}$ | $0.206_{\pm 0.008}$ | 239.889 | $0.556_{\pm 0.023}$ | $0.220_{\pm 0.005}$ | 253.737 |
| CA-in-GUARD | $0.206_{\pm 0.004}$ | $0.299_{\pm 0.004}$ | 168.356 | $0.667_{\pm 0.029}$ | $0.208_{\pm 0.007}$ | 236.523 | $0.558_{\pm 0.028}$ | $0.235_{\pm 0.005}$ | 243.897 |

*(b)* Best CLIP

| Method | SD v1.4 – verbatim memorization | | | SD v1.4 – template memorization | | | SD v2.0 – template memorization | | |
|---|---|---|---|---|---|---|---|---|---|
| | SSCD ($\downarrow$) | CLIP ($\uparrow$) | FID ($\downarrow$) | SSCD ($\downarrow$) | CLIP ($\uparrow$) | FID ($\downarrow$) | SSCD ($\downarrow$) | CLIP ($\uparrow$) | FID ($\downarrow$) |
| No mitigation | $0.875_{\pm 0.001}$ | $0.346_{\pm 0.001}$ | 243.056 | $0.776_{\pm 0.017}$ | $0.219_{\pm 0.007}$ | 258.976 | $0.735_{\pm 0.011}$ | $0.215_{\pm 0.005}$ | 303.266 |
| RTA | $0.328_{\pm 0.007}$ | $0.263_{\pm 0.002}$ | 175.866 | $0.617_{\pm 0.043}$ | $0.187_{\pm 0.010}$ | 218.343 | $0.543_{\pm 0.048}$ | $0.183_{\pm 0.009}$ | 233.580 |
| Wen et al. | $0.115_{\pm 0.011}$ | $0.267_{\pm 0.003}$ | 162.848 | $0.545_{\pm 0.038}$ | $0.188_{\pm 0.008}$ | 209.719 | $0.292_{\pm 0.027}$ | $0.191_{\pm 0.007}$ | 197.186 |
| Ren et al. | $0.142_{\pm 0.003}$ | $0.266_{\pm 0.004}$ | 163.670 | $0.602_{\pm 0.033}$ | $0.184_{\pm 0.007}$ | 222.066 | $0.356_{\pm 0.024}$ | $0.188_{\pm 0.007}$ | 208.416 |
| Han et al. | $0.191_{\pm 0.016}$ | $0.256_{\pm 0.008}$ | 166.551 | $0.479_{\pm 0.033}$ | $0.188_{\pm 0.006}$ | 210.839 | $0.401_{\pm 0.024}$ | $0.186_{\pm 0.005}$ | 208.852 |
| CA attenuation | $0.110_{\pm 0.007}$ | $0.284_{\pm 0.001}$ | 162.825 | $0.530_{\pm 0.038}$ | $0.185_{\pm 0.009}$ | 212.240 | $0.341_{\pm 0.021}$ | $0.204_{\pm 0.005}$ | 217.558 |
| CA-in-GUARD | $0.079_{\pm 0.007}$ | $0.266_{\pm 0.015}$ | 158.115 | $0.517_{\pm 0.038}$ | $0.186_{\pm 0.008}$ | 210.983 | $0.223_{\pm 0.017}$ | $0.183_{\pm 0.006}$ | 187.250 |

*(c)* Best FID

formance across most settings, achieving the strongest overall results for both SD v1.4 and SD v2.0. These results confirm that our methods reduce template memorization not only under SSCD, but also under an independent DINO-based retrieval evaluation.

### A.7.4. FULL-SPECTRUM EVALUATION

To examine whether our conclusions hold beyond the high-memorization subset used in the main evaluation in Section 6, we additionally evaluate all 500 examples from Webster (2023), providing a full-spectrum view across low-, medium-, and high-memorization cases. For each method and model architecture, we select the configuration that achieves the best SSCD score, since SSCD is our primary metric for memorization mitigation.

Table 7 reports SSCD, CLIP, and FID results across both SD v1.4 and SD v2.0 architectures. The results are consistent with our main findings: CA attenuation and CA-in-GUARD substantially reduce SSCD compared with the no-mitigation baseline, while maintaining competitive CLIP alignment and FID. In particular, CA-in-GUARD achieves the best SSCD and FID across both architectures, indicating that the proposed method reduces memorization without degrading overall generation quality. These full-spectrum results further support the robustness of our conclusions.

### A.7.5. COMPUTATIONAL EFFICIENCY

We compare the runtime and memory efficiency of our methods against all baselines, using wall-clock generation time per example and peak GPU memory usage under the default experimental setup described in Section 6. For each method, we measure the inference time and GPU memory consumption across both model architectures, and report the results in Table 8.

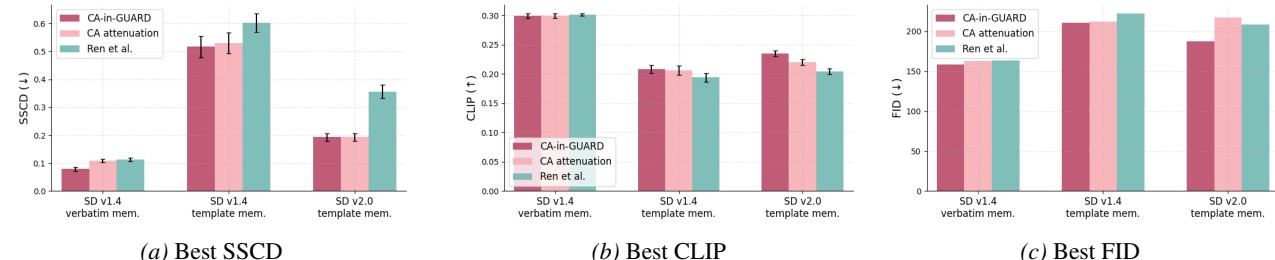

*(a)* Best SSCD          *(b)* Best CLIP          *(c)* Best FID

*Figure 12.* **The *best achievable* SSCD, CLIP, and FID of our methods compared to the prior state-of-the-art (and most related) baseline of Ren et al.**. We plot the best achievable value on each metric *individually*, using the configuration that yields best results on that metric. Because in each subplot, a different configuration may be used for a given method (the one that yields the best SSCD, best CLIP and best FID, respectively), this plot does not speak to the ability to do well on all metrics *jointly*, nor to trade-offs between these metrics, which we investigate later.

*Table 6.* SSCD and DINO results for template memorization settings, as an expansion of Table 1. Lower values indicate less similarity to the training set and therefore less memorization.

| Method | SD v1.4 – template memorization | | | SD v2.0 – template memorization | | |
| --- | --- | --- | --- | --- | --- | --- |
| | SSCD ($\downarrow$) | DINO@1 ($\downarrow$) | DINO@5 ($\downarrow$) | SSCD ($\downarrow$) | DINO@1 ($\downarrow$) | DINO@5 ($\downarrow$) |
| No mitigation | 0.776±0.017 | 0.924±0.006 | 0.906±0.013 | 0.735±0.011 | 0.903±0.006 | 0.895±0.007 |
| RTA | 0.617±0.043 | 0.836±0.030 | 0.813±0.035 | 0.543±0.048 | 0.794±0.038 | 0.778±0.039 |
| Wen et al. | 0.545±0.038 | 0.823±0.024 | 0.790±0.028 | 0.260±0.026 | 0.655±0.023 | 0.637±0.024 |
| Ren et al. | 0.602±0.033 | 0.858±0.015 | 0.832±0.020 | 0.356±0.024 | 0.711±0.020 | 0.695±0.020 |
| Han et al. | 0.479±0.033 | 0.832±0.015 | 0.789±0.019 | 0.401±0.024 | 0.745±0.017 | 0.732±0.018 |
| CA attenuation | 0.530±0.038 | 0.811±0.024 | 0.774±0.029 | 0.193±0.014 | 0.647±0.016 | 0.626±0.017 |
| CA-in-GUARD | 0.517±0.038 | 0.782±0.023 | 0.752±0.026 | 0.193±0.014 | 0.647±0.016 | 0.626±0.017 |

Overall, CA attenuation incurs minimal overhead, as it only manipulates cross-attention on-the-fly during generation. CA-in-GUARD is moderately more expensive due to the additional guidance components, but its memory usage remains close to the default generation setting and substantially lower than methods with heavier optimization or adjustment procedures. Notably, the implementation strategy described in Section A.2 allows the required forward passes to be batched efficiently, which could substantially reduce runtime overhead.

From a wall-clock perspective, the methods can be ranked from fastest to slowest as follows: RTA, CA attenuation, Wen et al. (2024), CA-in-GUARD, Han et al. (2025), and Ren et al. (2024). In terms of GPU memory usage, CA attenuation is comparable to the no-mitigation setting, while CA-in-GUARD remains lightweight relative to most baselines. These results indicate that our approaches achieve strong memorization mitigation while maintaining competitive inference and memory efficiency.

### A.8. Evaluation on SD v3.0

Our main experiments follow prior work and evaluate memorization mitigation on SD v1.4 and SD v2.0. We additionally evaluate SD v3.0 to test whether the empirical gains of CA-in-GUARD extend to a newer and substantially different architecture. Unlike SD v1.4 and SD v2.0, which use U-Net-based denoising networks, SD v3.0 uses a transformer-based backbone. As a result, some prior methods and some parts of our mechanistic analysis do not transfer directly. In particular, Ren et al. (2024) is not applicable in this setting and is therefore excluded.

We also note that the memorized prompt sets used in prior work were constructed for earlier Stable Diffusion models. Consequently, SD v3.0 exhibits substantially weaker memorization on this benchmark: only 39 of the 500 prompts satisfy SSCD > 0.3. We therefore evaluate SD v3.0 on this subset.

Table 9 reports the results. CA-in-GUARD achieves the lowest SSCD among all applicable methods, while also obtaining the best FID and comparable CLIP. CA attenuation is also competitive: although its SSCD is slightly higher than Han et al. (2025), it achieves better CLIP and FID, indicating a favorable overall trade-off. These results suggest that the empirical benefits of our approach extend beyond the standard SD v1.4/v2.0 evaluation setting. However, because memorization may arise differently in transformer-based diffusion models, we leave a more systematic study of memorization mechanisms and mitigation strategies for SD v3.0-style architectures to future work.

*Table 7.* Full-spectrum evaluation results across metrics and architectures. For each setting, we select for each method the configuration that yields the best SSCD, since SSCD is the primary metric for memorization mitigation.

| Method | SD v1.4 | | | SD v2.0 | | |
|---|---|---|---|---|---|---|
| | SSCD ($\downarrow$) | CLIP ($\uparrow$) | FID ($\downarrow$) | SSCD ($\downarrow$) | CLIP ($\uparrow$) | FID ($\downarrow$) |
| No mitigation | 0.524±0.028 | 0.257±0.006 | 198.081 | 0.333±0.025 | 0.260±0.005 | 146.527 |
| RTA | 0.366±0.026 | 0.223±0.007 | 153.361 | 0.274±0.023 | 0.223±0.006 | 135.615 |
| Wen et al. | 0.255±0.019 | 0.220±0.006 | 137.464 | 0.152±0.011 | 0.219±0.006 | 126.339 |
| Ren et al. | 0.350±0.024 | 0.223±0.006 | 144.830 | 0.200±0.016 | 0.218±0.006 | 130.757 |
| Han et al. | 0.239±0.018 | 0.220±0.006 | 143.297 | 0.176±0.023 | 0.221±0.006 | 125.894 |
| CA attenuation | 0.242±0.018 | 0.220±0.007 | 140.381 | 0.153±0.014 | 0.219±0.006 | 128.503 |
| CA-in-GUARD | 0.201±0.014 | 0.221±0.007 | 135.608 | 0.137±0.009 | 0.222±0.006 | 125.402 |

*Table 8.* Average wall-clock runtime per example (in seconds) and GPU memory usage (in GiB) of each method across both model architectures.

| Method | SD v1.4 | | SD v2.0 | |
|---|---|---|---|---|
| | Time (s) | Memory (GiB) | Time (s) | Memory (GiB) |
| No mitigation | 7.575 | 6.172 | 6.880 | 6.580 |
| RTA | 7.563 | 6.172 | 6.884 | 6.590 |
| Wen et al. | 9.210 | 12.697 | 8.709 | 13.102 |
| Ren et al. | 15.302 | 6.213 | 10.125 | 6.695 |
| Han et al. | 12.107 | 14.373 | 10.901 | 14.785 |
| CA attenuation | 7.770 | 6.142 | 7.095 | 6.625 |
| CA-in-GUARD | 10.963 | 6.543 | 9.713 | 6.735 |

*Table 9.* SD v3.0 evaluation results, extending Table 1. For each method, we report the configuration that achieves the best SSCD.

| Method | SSCD ($\downarrow$) | CLIP ($\uparrow$) | FID ($\downarrow$) |
|---|---|---|---|
| No mitigation | 0.337±0.011 | 0.213±0.021 | 336.888 |
| RTA | 0.286±0.021 | 0.182±0.023 | 275.955 |
| Wen et al. | 0.303±0.031 | 0.182±0.022 | 373.299 |
| Han et al. | 0.200±0.037 | 0.182±0.022 | 219.990 |
| CA attenuation | 0.204±0.032 | 0.185±0.027 | 215.708 |
| CA-in-GUARD | 0.168±0.017 | 0.181±0.020 | 212.728 |

## A.9. Qualitative Examples

We provide additional qualitative examples across model architectures and memorization settings, as shown in Figure 14, to complement the quantitative results in Figure 5 in the main paper. These examples compare generations from the no-mitigation baseline, prior mitigation methods, CA attenuation, and CA-in-GUARD against the corresponding training images.

Across examples, the no-mitigation baseline often preserves substantial visual overlap with the training image, especially in layout, object identity, and distinctive local details. Prior mitigation methods reduce this similarity to varying degrees, reflecting different trade-offs between memorization mitigation and generation quality. CA-in-GUARD shows the most consistent visual trade-off: it produces generations that are visually distinct from the corresponding training images while preserving prompt alignment and overall image

quality. This trend is visible in both verbatim and template memorization settings. Notably, even under template memorization, which is harder to mitigate quantitatively, CA-in-GUARD still substantially moves the generated image away from the memorized training example while maintaining good generation quality.

These qualitative results, together with the SSCD, CLIP, FID, and DINO results reported above, provide additional evidence that CA-in-GUARD mitigates memorization without relying on severe degradation of generation quality.

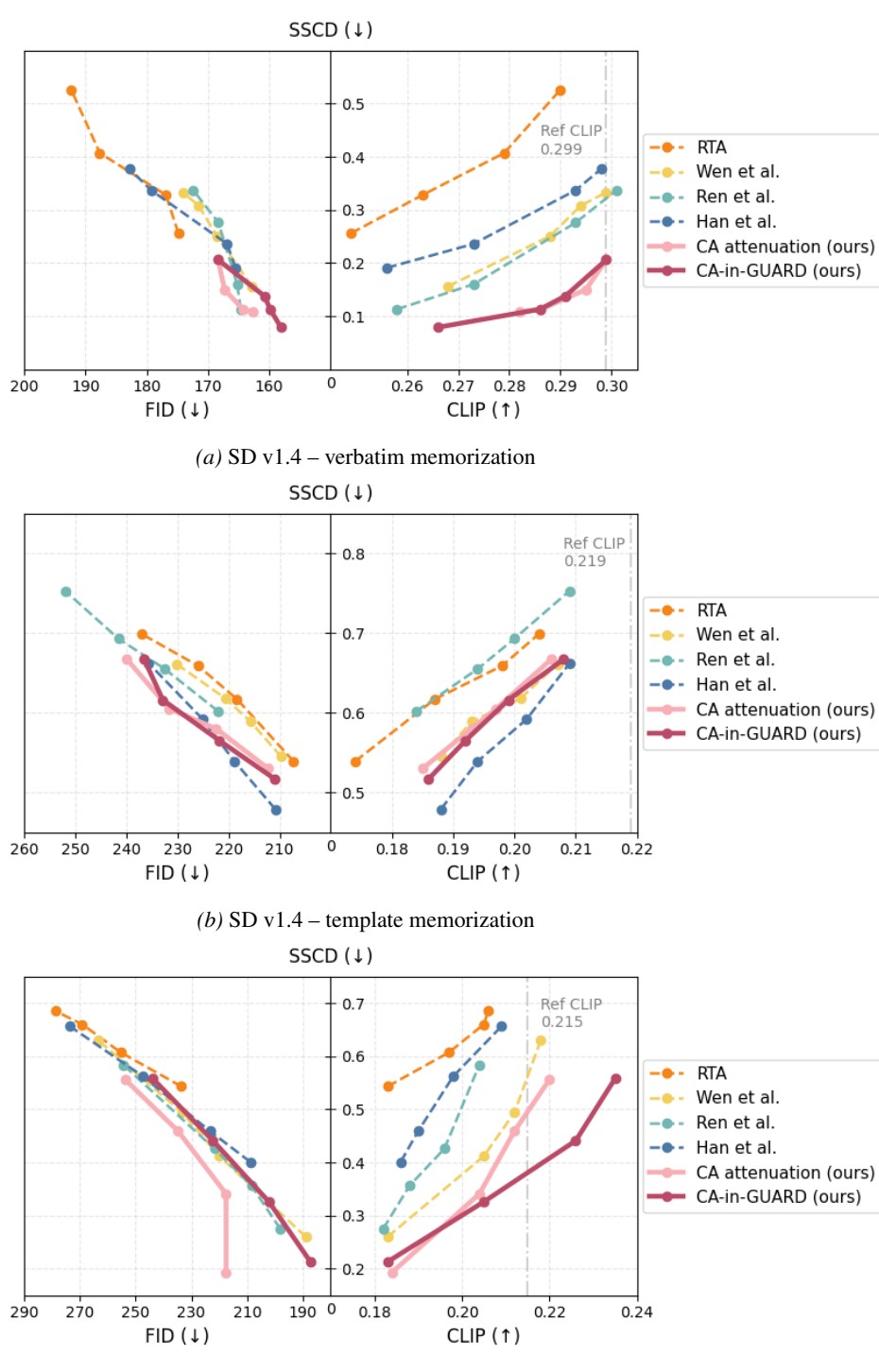

*(a)* SD v1.4 – verbatim memorization

*(b)* SD v1.4 – template memorization

*(c)* SD v2.0 – template memorization

*Figure 13.* **SSCD–FID and SSCD–CLIP Pareto frontiers comparing our methods with existing baselines under three memorization settings.** Lower SSCD and FID and higher CLIP indicate better performance; methods closer to the bottom-right corner represent more favorable trade-offs in both subfigures. The *reference CLIP score* indicates the target semantic fidelity that methods are expected to attain (see Section 6 – Hyperparameter selection for details on how this reference is defined).

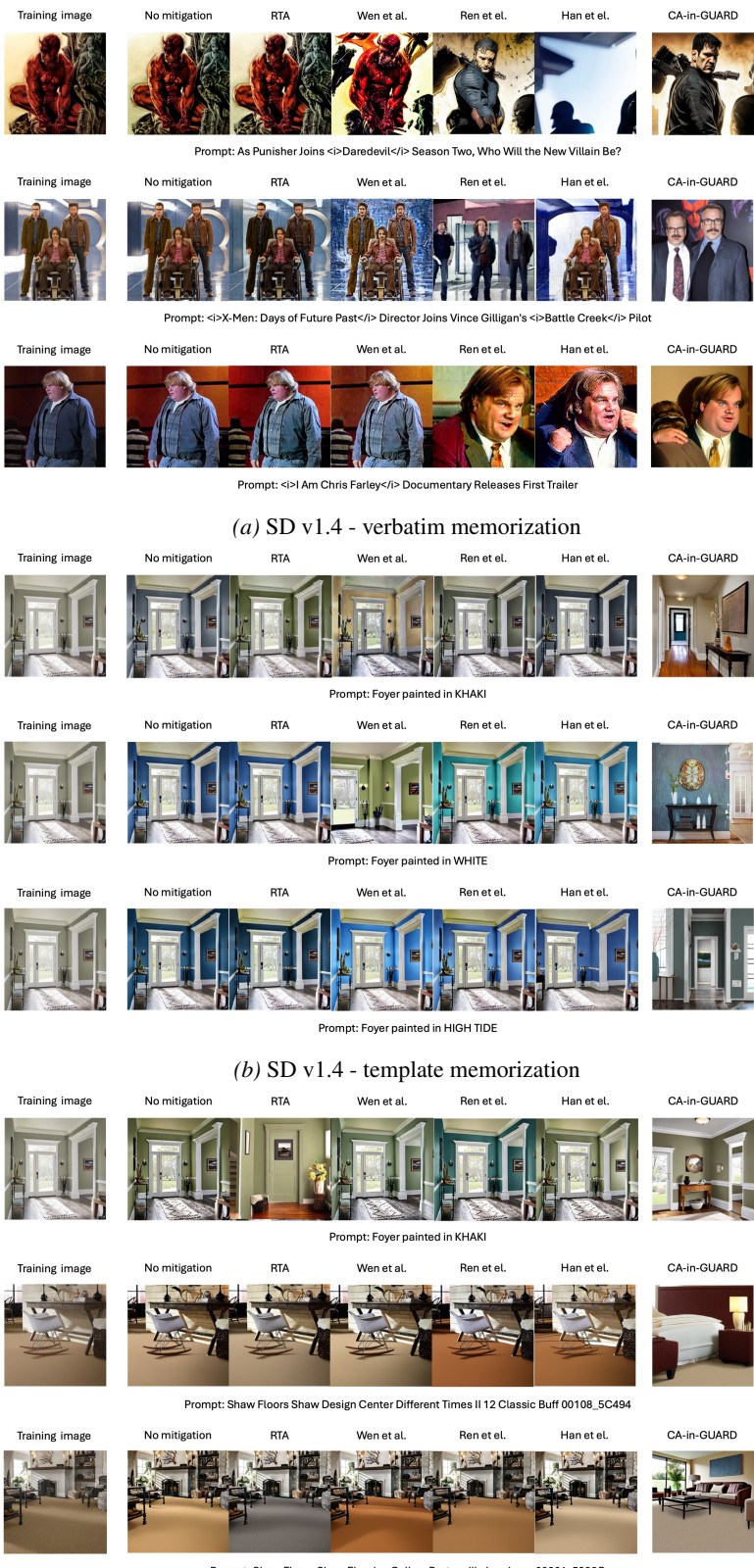

*(a)* SD v1.4 - verbatim memorization

*(b)* SD v1.4 - template memorization

*(c)* SD v2.0 - template memorization

*Figure 14.* Additional qualitative examples across model architectures and memorization settings. We compare the training image with generations from no mitigation, prior mitigation methods, CA attenuation, and CA-in-GUARD. CA-in-GUARD consistently moves generations away from memorized training images while preserving prompt-relevant visual content.

