# OpenReview forum: "You Don’t Need All That Attention: Surgical Memorization Mitigation in Text-to-Image Diffusion Models"
_ICML.cc/2026/Conference — ICML 2026 regular_

### Official Review · Reviewer_YHmm · 2026-03-02

**Soundness:** 3
**Presentation:** 3
**Significance:** 2
**Originality:** 3
**Overall Recommendation:** 4
**Confidence:** 4

**Summary:**

The paper addresses memorization in text-to-image diffusion models, where models may reproduce training images verbatim or template memorization, raising privacy and copyright concerns. It introduces GUARD, an inference-time framework that modifies the denoising guidance process by combining a repulsion term with an attraction term (to maintain image quality and prompt alignment). The authors further propose a dynamic, per-prompt cross-attention attenuation mechanism to detect and suppress memorization-critical tokens, and demonstrate that this approach outperforms prior methods in reducing memorization while preserving image quality.

**Compliance With Llm Reviewing Policy:**

Affirmed.

**Final Justification:**

The rebuttal addressed most of my concerns. I am thus raising my score.

**Key Questions For Authors:**

See weaknesses. My main concerns are the additional inference-time overhead introduced by the method, its generalizability to diffusion transformer architectures, and the justification for why redistributing the cross-attention distribution is expected to effectively mitigate memorization.

**Limitations:**

Yes

**Strengths And Weaknesses:**

**Strengths:**

* The paper is clearly written and generally easy to follow. The method and experimental setup are described in a structured and coherent manner.

* The idea of detecting cross-attention spikes and steering the model by rebalancing the cross-attention distribution is interesting and appears novel. The combination of spike detection with guidance-based steering is a compelling direction for inference-time memorization mitigation.

* The method is evaluated extensively across multiple settings and achieves comparable or slightly better performance than prior approaches overall. The improvements are particularly strong for verbatim memorization.

* The approach is training-free and operates purely at inference time, which is a practical advantage compared to training-time or finetuning-based methods.

**Weaknesses:**

* The method requires an additional forward pass for the positive target, storing cross-attention maps, and performing spike detection during inference. This likely increases generation time and memory usage. Since the approach requires explicit access to cross-attention maps, it may not be compatible with optimized implementations such as FlashAttention, potentially making it slower than current baselines. A quantitative comparison of runtime and memory consumption is required.

* The authors state (L138–140) that verbatim and template memorization arise from different mechanisms and exhibit distinct behavioral signatures. The basis for this claim is not clearly justified. It would help to clarify whether this statement is supported by theoretical arguments or prior empirical findings.

* The central assumption is that cross-attention spikes are related to memorization. However, the paper does not provide strong theoretical justification or convincing empirical evidence demonstrating a causal link, rather than mere correlation. The motivation for using spike detection as the core mechanism would benefit from deeper analysis or additional experiments.

* In Equation 5, there appear to be two unconditional noise predictions, ( \epsilon_\theta(x_t, e_\Phi) ). It is not clear why both are required. A more detailed explanation of this formulation would improve clarity.

* The setup behind the results shown in Figure 1 is not sufficiently clear. It would be helpful to specify which benchmark was used, how template memorization was defined and evaluated, and the exact experimental protocol used to produce these results.

* The experiments do not include transformer-based diffusion models, which are increasingly common and often achieve higher image quality. Evaluating the method on such architectures would make the empirical claims more convincing.

* It is unclear whether the proposed approach can be extended to other diffusion-based generative models, such as text-to-video or text-to-3D. Since these models also rely on cross-attention mechanisms, a discussion on generalization would strengthen the paper.

* Figure 1 is somewhat difficult to follow. The text alternates between panels (a) and (c), and then (b) and (d), which makes it harder to read. Grouping corresponding panels together and increasing the font size would improve clarity.

* Although prior work is cited suggesting that memorization is associated with large attention concentration on specific “trigger tokens,” the motivation for analyzing particular tokens (e.g., EOT) is not sufficiently explained. It would help to clarify whether this choice is directly grounded in previous findings and to better articulate the reasoning behind this analysis.

---

> ### Author Rebuttal · Authors · 2026-03-31
>
> We thank the reviewers for the feedback and address all comments below.
>
> **W1**: We would like to clarify that all forward passes in GUARD are batched into a single forward pass (Appendix A.2), so no additional forward pass is needed. The FlashAttention concern is reasonable as such implementations avoid materializing full attention maps, but it applies equally to **all** baselines that intervene on cross-attention maps (e.g., Ren et al.). More importantly, standard SD inference pipelines do not use FlashAttention by default, so our reported runtimes still reflect the practical and commonly used deployment setting. To address the concern, we added both runtime and memory comparisons (see Reviewer yyBi - W5), both of which confirm that GUARD remains efficient and comparable to SOTA. Extending GUARD to FlashAttention-compatible implementations would be an interesting engineering direction for future work.
>
> **W2**: This claim is grounded in prior work: [Webster et al.] showed deduplication substantially reduces verbatim memorization, and [Ren et al.] observed that SD v2.0 – trained on a deduplicated dataset – exhibits little to no verbatim memorization. Together, these support the view that the two types of memorization have distinct origins. We will make these citations and reasoning more explicit in the revision.
>
> **W3**: Our claim is not that we establish a full causal theory of memorization, but that spike-based cross-attention signals are a practically useful and empirically supported intervention point. This motivation is grounded in [Ren et al], and we generalizes this to a broader guidance framework, yielding more flexibility and substantially stronger performance across settings. We refer the reviewer to Sec. 3 - ITMM for a detailed discussion.
>
> **W4**: The two occurrences of ε_θ​(x_t​,e_ϕ​) refer to the same unconditional prediction, reused as a shared reference point for two guidance directions: the “attraction” term defines attraction toward the positive target; the “repulsion” term defines repulsion away from the memorized target. This directly mirrors standard CFG, where conditional guidance is also defined relative to the unconditional prediction. We will revise the text to make this clearer.
>
> **W5**: Following [Webster; Wen et al]: verbatim memorization refers to one-to-one (prompt, image) pairs from the training data, where a specific prompt reliably reproduces a specific training image; template memorization refers to one-to-many/many-to-many (prompt, image) pairs, where broader patterns are reproduced across varied prompts or outputs. The remaining experimental details are described in Sec. 6. We will add these definitions to the main text and add a concise summary to the Fig. 1 caption.
>
> **W6**: Our evaluation follows the standard protocol established by all prior work in this area, focused on UNet-based SD architectures where memorized prompts are systematically benchmarked. It is also not yet well established to what extent memorization manifests in newer transformer-based architectures, so the suggested setting itself requires additional investigation. However, to address the concern, we have added SD v3.0 results (see Reviewer GJqn - W1) -- to our knowledge, the first memorization mitigation evaluation in transformer-based SD architecture -- confirming CA-in-GUARD significantly outperforms beyond UNet-based architectures. A more systematic study of memorization and its mitigation in this setting is an important direction we leave for future work.
>
> **W7**: GUARD's core principle, i.e. detecting abnormal cross-attention behavior and intervening on the guidance signal, is not inherently limited to T2I diffusion.
>
> For text-to-video, most diffusion-based models retain cross-attention-based text conditioning [6], so spike detection and repulsive guidance could plausibly extend there, though temporal consistency across frames in video would likely require joint rather than per-frame spike regularization.
>
> For text-to-3D, applicability depends on whether the pipeline is 2D-guided (SDS-based) or 3D-native, which also uses text-guided attention mechanisms or explicit cross-attention during generation. This suggests that our core idea may also transfer there, but the relevant notion of memorization and the appropriate intervention point will depend on specific cases. We are excited to see future work explore these directions, though they are beyond the scope of this paper.
>
> **W8**: Thanks for the feedback. We will reorganize Figure 1 to group corresponding panels together and increase the font size to improve readability.
>
> **W9**: We respectfully direct the reviewer to Sec.3 - ITMM, where we discuss the motivation for our EOT-focused analysis in detail, grounding it explicitly in the empirical findings of [Ren et al.]. We believe the current manuscript already addresses this comprehensively, but we are happy to provide further clarification if any specific aspect remains unclear.

---

> > ### Author Rebuttal · Reviewer_YHmm · 2026-04-03
> >
> > Thank you for the detailed rebuttal, which addressed most of my concerns. I am thus raising my score.

---

### Official Review · Reviewer_GJqn · 2026-03-09

**Soundness:** 3
**Presentation:** 3
**Significance:** 3
**Originality:** 3
**Overall Recommendation:** 4
**Confidence:** 3

**Summary:**

This paper proposes a method called GUARD to address the "memorization" of training data in generative models, thereby mitigating risks of privacy leakage and copyright infringement without requiring any retraining. By employing an attention-spike detector based on finding statistical outliers in the Cross-Attention (CA) distribution, the method automatically identifies attention positions within each prompt that require attenuation, thus enabling precise control over memorized tokens.

**Compliance With Llm Reviewing Policy:**

Affirmed.

**Final Justification:**

The author's further response completely resolved my issue, so I will keep the score as weak accepted.

**Key Questions For Authors:**

1. In Table 1, under "SD v2.0 – template memorization," the SSCD scores for CA attenuation and CA-in-GUARD are exactly identical. Does this indicate that introducing a negative target yields no improvement on stronger baselines?
2. In Table 1, under "SD v1.4 – template memorization," neither CA attenuation nor CA-in-GUARD achieved SOTA, as their SSCD scores were both higher than those of the Han et al. method. What is the reason for this?
3. Since CA-in-GUARD modifies the calculation formula for the model's CFG, it would be interesting to discuss its impact under different CFG scales, such as ranging from 1 to 7.

**Limitations:**

yes

**Strengths And Weaknesses:**

Strengths :
- GUARD is plug-and-play, requiring no retraining or modification of the generative model's weights.
- Compared to existing methods, GUARD significantly reduces memorization behavior (achieving lower SSCD scores) without causing excessive degradation in image quality.

Weaknesses：
- This method was only tested on SDv1.4 and SDv2.0. Why didn't the authors test latest models such as SDv3.0 and FLUX.2-dev?
- Additional inference overhead: During the inference phase, CA-in-GUARD requires computing image generation results for the Negative target in addition to the Positive target and the Unconditional input.

---

> ### Author Rebuttal · Authors · 2026-03-31
>
> We thank the reviewers for their feedback and address all comments below.
>
> **W1**: We add new experiments for SD v3.0 (Table B). SD v3.0 uses a transformer-based architecture rather than U-Net, so Ren et al. is not applicable and is excluded. We also note that existing memorized prompt sets used across all prior work were established for earlier SD models [Webster et al], so only 39 of the 500 prompts satisfy SSCD > 0.3 on SD v3.0, and we evaluated on these. Also, we would like to emphasize that prior work evaluates primarily on SD v1.4/v2.0; **our new SD v3.0 results already go beyond the standard evaluation setting and to our knowledge, present the first evaluation of any memorization mitigation methods on SD v3.0**. Table B shows that CA-in-GUARD significantly outperforms all SOTAs, suggesting our gains are robust across substantially different architectures.
>
> Table B. SD v3.0 evaluation results (best SSCD configuration per method).
> | Method | SSCD | CLIP | FID |
> |---|---|---|---|
> | No mitigation | 0.337±0.011 | 0.213±0.021 | 336.888 |
> | RTA  | 0.286±0.021 | 0.182±0.023 | 275.955 |
> | Wen et al. | 0.303±0.031 | 0.182±0.022 | 373.299 |
> | Han et al. | 0.200±0.037 | 0.182±0.022 | 219.990 |
> | CA attenuation | 0.204±0.032 | 0.185±0.027 | 215.708 |
> | CA-in-GUARD | 0.168±0.017 | 0.181±0.020 | 212.728 |
>
> **W2**: CA-in-GUARD's runtime is comparable to prior SOTA baselines, **all of which also introduce overhead** relative to no mitigation (Appendix A.8.3). We additionally report GPU memory usage and per-example runtime across all methods in Table C at https://anonymous.4open.science/r/rebuttal-guard-4C2A, confirming CA-in-GUARD is highly efficient and comparable to existing baselines. Moreover, all inference-time methods, including GUARD, require no retraining/fine-tuning, which is a major advantage over training-based alternatives. Therefore, the overhead is justified given the substantially stronger mitigation performance.
>
> **Q1**: As discussed in Sec. 5.2, CA attenuation is a special case of CA-in-GUARD with repulsive scale r=0. Importantly, hyperparameters are tuned separately for each setting, and in the SD v2.0 - template memorization setting, the best-performing configuration happens to be r=0, hence identical SSCD scores. But this does **not** mean that the negative target provides no benefit: across the other two settings in Table 1, **CA-in-GUARD achieves substantially better SSCD** than CA attenuation alone; and even in the SD v2.0 - template mem. setting where the SSCD values are identical, **CA-in-GUARD achieves better CLIP and substantially better FID**, indicating a better overall trade-off.
>
> **Q2**: Han et al. outperforms our method on SSCD in only **one specific setting** (SD v1.4 - template memorization); it underperforms substantially in the other settings (Tables 1,4,B), often not even ranking as the second-best method, whereas CA-in-GUARD remains consistently strong. Moreover, under the added **DINO retrieval** evaluation (refer to Reviewer yyBi-W7, Table D at the link), our methods outperform Han et al. even in that same setting, despite Han et al. obtaining a lower SSCD there. This suggests the SSCD difference in this one case does not reflect a broader pattern. Han et al. is also substantially more computationally expensive (Appendix A.8.3). Overall, it is clear from Tables 1,4,A,B / Figures 2,3 that CA-in-GUARD offers by far the best overall balance of memorization mitigation, generation quality, cross-setting robustness, and efficiency.
>
> **Q3**: Following this suggestion, we add ablations for CFG scales s∈{1,4,7,7.5} (default: 7.5), reporting both CA-in-GUARD and the no-mitigation baseline at each (Figure B-d at W2’s link).
>
> Key findings: (i) Without mitigation, lower s reduces memorization (SSCD 0.875 --> 0.571) but degrades generation quality. This is expected according to the standard CFG formula (eq.(3)): ϵ_pred = ϵ_uncond + s(ϵ_cond − ϵ_uncond), where s controls how strongly sampling is steered toward the prompt-conditioned direction. Lower s weakens prompt alignment, which reduces memorization but hurts fidelity.
>
> (ii) CA-in-GUARD remains effective across all scales, consistently reducing SSCD relative to the no-mitigation baseline while maintaining a favorable quality trade-off.
>
> (iii) At s=1, mitigation performance drops, but not specific to GUARD: e.g. the strong baseline of Wen et al. shows only limited (and worse than CA-in GUARD) SSCD improvement over the “no mitigation” baseline (Table B-d at the link).
> Our interpretation: larger s amplified the prompt-conditioned direction, increasing both memorization risk and the structured guidance signal that mitigation methods can intervene on. At s=1, CFG reduces to plain conditional denoising, leaving less headroom for any inference-time method to operate on (because most inference-time methods operate by modifying the guidance signal or its induced trajectory). We leave a more systematic study of this interaction as future work.

---

> > ### Author Rebuttal · Reviewer_GJqn · 2026-04-02
> >
> > Thanks for your detailed responses. My concerns are almost addressed.

---

> > > ### Author Response · Authors · 2026-04-05
> > >
> > > Dear reviewer GJqn,
> > >
> > > Thank you for your engagement and rebuttal acknowledgement.
> > > We are very encouraged that our rebuttal addressed all of your concerns.
> > > Given this, we would kindly ask if you might consider raising your score to reflect this fact. We believe this would assist the Area Chair in evaluating the paper's final standing.
> > >
> > > Thank you again for your time and consideration.
> > >
> > > Best regards,
> > >
> > > The Authors

---

### Official Review · Reviewer_yyBi · 2026-03-11

**Soundness:** 3
**Presentation:** 3
**Significance:** 2
**Originality:** 2
**Overall Recommendation:** 4
**Confidence:** 4

**Summary:**

This paper addresses example-level memorization in text-to-image diffusion models and proposes GUARD, an inference-time framework that modifies classifier-free guidance by jointly repelling from the standard text-conditional signal and attracting toward a “positive target” conditional signal. The authors instantiate the positive target via a per-prompt, per-step cross-attention spike detector that attenuates logits at statistically identified trigger tokens, yielding a dynamic and surgical mitigation. Across Stable Diffusion v1.4/v2.0 and both verbatim and template memorization, the method (especially CA-in-GUARD) reduces SSCD substantially while preserving or improving CLIP/FID compared to strong baselines.

**Compliance With Llm Reviewing Policy:**

Affirmed.

**Final Justification:**

The author responses addressed all my concerns, I hence increase my score to 4.

**Key Questions For Authors:**

1. How does the method interact with negative prompts, prompt-augmentation tricks, or LoRA/style adapters? Does the spike detector remain reliable when adapters alter attention distributions?
2. Could you add evaluations on newer backbones (e.g., SDXL-base/refiner) to assess generality? If not feasible, do you have preliminary evidence that CA-in-GUARD transfers without re-tuning?
3. For template memorization, do you have retrieval-based analyses (e.g., CLIP/DINO nearest neighbors in the pretraining set) to verify that reductions in SSCD correspond to reduced proximity to the specific training items rather than generic style shifts?
4. Can you report concrete efficiency numbers (latency, memory footprint, throughput) for CA attenuation and CA-in-GUARD vs. vanilla CFG and vs. the strongest baselines, across SD v1.4/v2.0 and typical batch sizes/schedulers?
5. How does GUARD behave under different samplers (DDIM, Euler a, Heun, DPM++), guidance schedules, and step counts? Are there regimes where repulsion harms stability or semantic alignment?

**Limitations:**

Risks remain: inference-time defenses can be bypassed under white-box access or by changing the sampling stack; and the long-term desideratum of removing memorized content from weights remains. Still, for many providers operating under black-box access models, the method provides meaningful risk reduction at low cost.

**Strengths And Weaknesses:**

## Strengths
**Experimental rigor and validation**
- Comprehensive comparisons against several recent inference-time methods (RTA, Wen et al., Ren et al., Han et al.) on two architectures and two memorization regimes.
- Ablation studies on: positive target design (semantic paraphrases vs. CA attenuation), which blocks/heads/timesteps to attenuate, and effects on non-memorized prompts.
- Clear reporting of SSCD (memorization) and CLIP/FID (quality) with confidence intervals, and exploration of trade-offs.

**Clarity of presentation**
- The GUARD formulation and its relationship to standard CFG are clearly laid out, with an intuitive interpretation of repulsion and attraction.
- The cross-attention spike-detection/attenuation procedure is described at an appropriate level of detail, including where in the U-Net - and when across timesteps it is applied.
- The paper explains why EOT-only attenuation can harm template-memorization performance and supports this with attention-distribution analyses.

## Weaknesses
**Technical limitations or concerns**
- Inference-time mitigation cannot remove memorized information from model weights and may be circumvented under white-box access; while acknowledged, a more explicit discussion of realistic attacker capabilities in black-box vs. gray-box settings would strengthen the threat-model framing.
- The GUARD update can be algebraically rewritten to show its relation to conventional CFG with an additional delta term; a deeper theoretical analysis of stability and convergence under typical samplers would add confidence.

**Missing related work or comparisons**
- Discussion of broader CFG variants (e.g., negative prompts, guidance mixing strategies) and how GUARD relates to or subsumes them could be expanded.
- Additional recent work on attention manipulation and scheduler effects in diffusion sampling might be acknowledged to contextualize CA-in-GUARD’s design choices.

**Experimental gaps or methodological issues**
- Efficiency claims rely on batching and qualitative statements; more concrete wall-clock, memory, and throughput benchmarks across batch sizes/schedulers would be helpful.
- Evaluation is limited to SD v1.4 and v2.0; results on SDXL or other recent T2I backbones would improve generality claims.
- Metrics: SSCD is appropriate for verbatim copying, but template memorization is more nuanced; additional retrieval-style or perceptual metrics (e.g., DINO/CLIP retrieval to training set neighborhoods) or human assessment could strengthen conclusions for template cases.
- The prompt filtering to SSCD > 0.7 focuses on “hard” cases but potentially biases comparisons; full-spectrum results are mentioned but not foregrounded.

---

> ### Author Rebuttal · Authors · 2026-03-31
>
> We thank the reviewer for their thorough engagement. We have replied to every comment the reviewer made and run a large number of experiments. In summary, our method (i) significantly outperforms prior SOTA on SD v3.0; (ii) works across samplers, step counts, and schedules; (iii) is compute- and memory-efficient; (iv) remains top-performing under a retrieval-based metric; (v) remains top-performing under full-spectrum evaluation. Based on all of these findings, we believe the reviewer should be convinced that without doubt, GUARD is a solid contribution and substantially pushes the state-of-the-art on a realistic and important problem setting. We respond point-by-point below:
>
> **W1, Limitations**: We will clarify the attacker model in the revision. GUARD targets the black-box deployment setting, where the attacker queries the model via its public interface to extract memorized content without access to model internals, and inference-time mitigation is directly relevant because it reduces the probability of regurgitation at the point of output generation. This is a **realistic scenario**: for non-open-source models, it’s reasonable to assume that the attacker has query access but not access to the model’s internals. By contrast, in gray/white-box settings, attackers can access model internals or control inference.
>
> **W2**: The GUARD update is indeed expressible as a standard CFG with a delta term – we present it this way in the paper and view it as a feature that enables easy integration into existing pipelines. Since GUARD does not introduce a new optimization procedure, we are not sure what "stability" or "convergence" refers to here. We are happy to clarify if the reviewer has a more specific concern.
>
> **W3, Q1**: We will expand our discussion of related inference-time guidance methods. GUARD differs from negative prompting and many prompt-augmentation tricks, which require specifying problematic content in advance. Guidance-mixing strategies are related at a high level as inference-time interventions but are general-purpose; GUARD instead introduces a targeted correction specifically designed to steer sampling away from memorized directions toward a safer output, without assuming the problematic content can be specified beforehand as a semantic concept. Re: LoRA/style adapters: we adopt the deployment-time setting where the model is fixed, which is the most common use case. Adapter interactions are an interesting extension but beyond our scope, and to our knowledge has not been studied in prior memorization mitigation work. That said, our spike detector is based on a relative, per-prompt criterion (not a fixed threshold), so it should be naturally more robust to moderate shifts in attention scale.
>
> **W4**: Related attention-based methods are discussed in Section 3 - ITMM. On scheduler effects, our paper already includes scheduler-related ablations for CA attenuation in A.7; following the reviewer’s suggestion, we now add the corresponding ablations for CA-in-GUARD as well (see Figure B at: https://anonymous.4open.science/r/rebuttal-guard-4C2A). Results show that CA-in-GUARD is effective across all scheduler variants tested.
>
> **W5, Q4**: We report wall-clock runtimes in Appendix A.8.3, and to further address reviewer’s concern, we additionally include GPU memory usage and per-generation runtimes in Table C (see link in W4). These results confirm that CA-in-GUARD remains efficient and practically comparable to prior SOTA baselines.
>
> **W6, Q2**: We added experiments on SD v3.0, and GUARD continues to substantially outperform prior methods there as well, see our response to Reviewer GJqn - W1
>
> **W7, Q3**: We agree this is useful. While SSCD is already a perceptual similarity metric widely used for both verbatim and near-verbatim memorization in prior work, we additionally evaluated DINO retrieval to training-set neighborhoods for both template-memorization settings and reported top-1 and top-5 retrieval results in Table D (at the link in W4). The conclusions are consistent with SSCD: our CA and CA-in-GUARD remain top-performing across both SD v1.4 and SD v2.0.
>
> **W8**: We agree that showing results beyond the SSCD>0.7 subset gives a more complete picture. We therefore added evaluation over all 500 examples (see Table E at the link). The full-spectrum results are consistent with our main findings, which further support the robustness of our conclusions.
>
> **Q5**: For comparability, we follow prior work’s standard setup: DDIM, 50 steps, CFG 7.5. Following the reviewer’s suggestion, we additionally tested different samplers (DDIM, Euler a, Heun, DPM++), step counts (30, 50, 70), guidance schedules (none, cosine, linear) and CFG scales (1, 4, 7, 7.5), see Figure B at the link in W4. Overall, CA-in-GUARD remains robust and performs well across these settings. Please refer to our response to reviewer GJqn - Q3 for a detailed discussion on CFG scales.

---

> > ### Author Rebuttal · Reviewer_yyBi · 2026-04-01
> >
> > Thanks for your detailed responses. My concerns are almost addressed and I will increase my score accordingly.

---

### Official Review · Reviewer_P3A3 · 2026-03-12

**Soundness:** 2
**Presentation:** 2
**Significance:** 3
**Originality:** 2
**Overall Recommendation:** 3
**Confidence:** 4

**Summary:**

This paper aims to address the severe data memorization issue in text-to-image diffusion models, where the model generates training data either exactly or nearly unchanged, raising privacy and copyright concerns. The authors propose a mitigation framework called GUARD  which operates at inference time. This framework modifies classifier-free guidance (CFG) by introducing dual dynamics: "repulsion" (noise prediction that moves away from memory-triggering prompts) and "attraction" (noise prediction that moves toward safe targets). To construct high-quality safe targets, the paper analyzes the distribution patterns of cross-attention in depth and proposes a dynamic CA spike detection and attenuation mechanism . This mechanism identifies memory-triggering tokens in real-time for each prompt during inference using statistical methods and precisely weakens their attention weights. Experiments show that this approach significantly reduces the model's memorization effects (measured by the SSCD metric) while maintaining the quality of image generation.

**Compliance With Llm Reviewing Policy:**

Affirmed.

**Final Justification:**

I thank the authors for their rebuttal. After reviewing the qualitative cases provided, I remain skeptical about the effectiveness of the proposed method. Specifically, the reliance on SD 1.4—which is now a significantly outdated baseline—limits its impact. Considering its contribution to the community and practical utility, I will maintain my original score.

**Key Questions For Authors:**

Please see weakness.

**Limitations:**

yes

**Strengths And Weaknesses:**

**Strengths**:
1.  The paper provides a detailed distinction analysis between "verbatim memory" and "template memory." The author finds that although verbatim memory exhibits an extremely high CA spike at the end-of-text (EOT) token, template memory actually displays multiple widely distributed CA spikes, and its early EOT attention weight is even lower than that of non-memory samples. This finding is highly insightful.
2. This mechanism does not lead to a significant drop in generation quality when dealing with normal prompts that are not memorized. This relaxes the strict precondition of "having to pre-screen which prompts belong to the memorized samples" in actual deployment.

**Weakness**:
1. Extreme lack of qualitative visual presentation: the main text and appendix are filled with a large number of charts (such as Pareto Frontiers and CA quality distribution diagrams), yet there is not a single comparative display of actual generated images. Readers cannot visually evaluate the differences between the "training original images," "baseline memory-generated images," and "GUARD-generated images" on a visual level, nor can they intuitively perceive the extent of semantic preservation in the images after reducing CA.
2. Hyperparameter sensitivity and debugging costs: The GUARD framework and dynamic CA detection introduce several critical hyperparameters, including guidance strength, decay coefficient, and decision threshold. As shown in Appendix Table 2, the optimal search ranges for these hyperparameters vary significantly across different model architectures (SD v1.4 vs. v2.0) and memory types. In real-world black-box or unknown distribution scenarios, the cost of hyperparameter tuning is relatively high.

---

> ### Author Rebuttal · Authors · 2026-03-31
>
> We thank the reviewer for their careful reading and constructive feedback. We address each point below.
>
> **W1**: We agree that qualitative examples are important for visually assessing both memorization mitigation and semantic preservation. To address this concern, we have added comparative visual examples at: https://anonymous.4open.science/r/rebuttal-guard-4C2A
>
> The examples include: (i) the original training image, (ii) the image generated by the pretrained model with no mitigation, (iii) the image generated after applying previous SOTA methods, and (iv) the image generated after applying our method. These results show that **CA-in-GUARD reduces memorization while better preserving visual quality**. We will incorporate these examples into the revised manuscript to complement the quantitative results and make the memorization mitigation effect more directly visible to the reader.
>
> **W2**: We thank the reviewer for raising this important practical point. Our method introduces only three interpretable hyperparameters: the spike-detection threshold τ, the attenuation factor α, and for GUARD, the repulsive scale r. These parameters play distinct and intuitive roles: τ determines which unusually-dominant tokens are identified as spikes, α controls how strongly the detected spikes are attenuated, and r controls the strength of the repulsive guidance term of GUARD.
>
> We would like to clarify that the hyperparameter ranges reported in Appendix Table 2 reflect the full search space we explored for transparency, not a requirement to exhaustively tune all combinations in practice. In practice, our method behaves predictably: stronger memorization mitigation is generally achieved by increasing r and/or lowering τ and α. This interpretable and monotonic relationship makes the method straightforward to control without extensive tuning.
>
> Furthermore, our experiments demonstrate that GUARD does not rely on fragile hyperparameter choices to be effective, reasonable default values transfer well across evaluated settings (see Table A below). We will revise the manuscript to make this point explicit and to better convey the practical interpretability of these controls.
>
> Finally, we note that relative to SOTA baselines, GUARD introduces a modest number of hyperparameters. Most competing methods require comparable or greater tuning effort (with RTA as an exception, though it is substantially outperformed across all evaluations).
>
> Overall: GUARD does not have a larger number of hyperparameters than competing methods, and is shown not to be overly sensitive to hyperparameter choices.
>
> Table A. Default values of hyperparameters for CA attenuation and CA-in-GUARD across different settings.
>
> | Settings  | CA attenuation (τ) | CA attenuation (α) | CA-in-GUARD (τ) | CA-in-GUARD (α) | CA-in-GUARD (r) |
> |-----|-----|------|------|------|-----|
> | SD v1.4 - Verbatim mem.  | 0.01  | 0.3  | 0.1  | 0.3  | 2.0  |
> | SD v1.4 - Template mem.  | 1.5  | 0.5  | 1.5  | 0.7  | 2.0  |
> | SD v2.0 - Template mem.  | 1.5  | 0.5  | 1.5  | 0.7  | 2.0  |

---

> > ### Author Rebuttal · Reviewer_P3A3 · 2026-04-04
> >
> > please see my new comment

---

> > > ### Author Response · Authors · 2026-04-08
> > >
> > > Thank you for your comments. We would like to respectfully clarify a few points and provide additional evidence to address your concerns.
> > >
> > > First, our contribution is not limited to SD v1.4. The paper includes comprehensive quantitative results across SD v1.4 and SD v2.0, with extensive ablations. These settings are consistent with the standard evaluation framework used by all prior work in this area. Moreover, in our rebuttal we additionally provided new results on SD v3.0, which **already go beyond the standard evaluation setting and to our knowledge, present the first evaluation of any memorization mitigation methods on SD v3.0** (please refer to our response to **Reviewer GJqn - W1**). We therefore respectfully disagree that the contribution can be characterized as a "reliance on SD 1.4."
> > >
> > > Second, we have also updated our qualitative results to include visualisations across SD v1.4, SD v2.0, and SD v3.0, directly addressing the concern about practical relevance. Please find them at: https://anonymous.4open.science/r/rebuttal-guard-4C2A
> > >
> > > Finally, we would like to note that your additional comments were not visible to us until the final justification was posted, so we were unable to respond earlier. Nonetheless, we have done our best to address all raised concerns, and we believe they have been fully resolved. We sincerely hope you will consider revising your score in light of the discussion.

---

### Decision · Program_Chairs · 2026-04-30

**Decision:**

Accept (regular)

**Comment:**

The paper addresses an important and timely problem of mitigating memorization in text-to-image diffusion models. It proposes a practical inference-time framework that combines a well-motivated guidance formulation with a dynamic cross-attention intervention. Reviewers found the method technically solid, clearly presented, and extensively evaluated, with strong empirical evidence that it substantially reduces memorization while largely preserving generation quality across multiple settings. The rebuttal further strengthened the paper by clarifying the threat model, efficiency, and generalization, and by adding results on newer architectures and broader evaluation protocols, which resolved most reviewer concerns. Although some limitations remain, particularly regarding breadth of validation and practical deployment details, the consensus is that the paper makes a meaningful contribution to an important problem and is likely to be of interest to the community.